# Linker histone H1.8 inhibits chromatin binding of condensins and DNA topoisomerase II to tune chromosome length and individualization

**Pavan Choppakatla[1], Bastiaan Dekker[2], Erin E Cutts[3], Alessandro Vannini[3,4], Job Dekker[2,5], Hironori Funabiki[1]\***

[1]Laboratory of Chromosome and Cell Biology, The Rockefeller University, New York, United States; [2]Program in Systems Biology, Department of Biochemistry and Molecular Pharmacology, University of Massachusetts Medical School, Worcester, United States; [3]Division of Structural Biology, The Institute of Cancer Research, London, United Kingdom; [4]Fondazione Human Technopole, Structural Biology Research Centre, 20157, Milan, Italy; [5]Howard Hughes Medical Institute, Chevy Chase, United States

**Abstract** DNA loop extrusion by condensins and decatenation by DNA topoisomerase II (topo II) are thought to drive mitotic chromosome compaction and individualization. Here, we reveal that the linker histone H1.8 antagonizes condensins and topo II to shape mitotic chromosome organization. In vitro chromatin reconstitution experiments demonstrate that H1.8 inhibits binding of condensins and topo II to nucleosome arrays. Accordingly, H1.8 depletion in *Xenopus* egg extracts increased condensins and topo II levels on mitotic chromatin. Chromosome morphology and Hi-C analyses suggest that H1.8 depletion makes chromosomes thinner and longer through shortening the average loop size and reducing the DNA amount in each layer of mitotic loops. Furthermore, excess loading of condensins and topo II to chromosomes by H1.8 depletion causes hyper-chromosome individualization and dispersion. We propose that condensins and topo II are essential for chromosome individualization, but their functions are tuned by the linker histone to keep chromosomes together until anaphase.

\*For correspondence:
funabih@rockefeller.edu

## Introduction

Genomic DNA in eukaryotes is compacted by orders of magnitude over its linear length. The extent and mode of the packaging change between interphase and mitosis to support the cell cycle-dependent functions of the DNA. While loosely packed DNA allows efficient decoding of genetic information in interphase, mitotic compaction of DNA enables efficient distribution of genetic information to daughter cells. In addition, chromosome individualization during mitosis ensures that all duplicated chromosomes are independently moved by microtubules and are equally distributed to daughter cells. Despite these common functional requirements of mitotic chromosomes and evolutionary conservation of major known regulators of mitotic chromosome structures, the size and shape of chromosomes vary among species and developmental stages. For example, rod-like individual mitotic chromosomes can be readily visualized in fission yeast *Schizosaccharomyces pombe*, but not in budding yeast *Saccharomyces cerevisiae* (*Guacci et al., 1994*; *Umesono et al., 1983*). During early embryogenesis in *Xenopus* and *Caenorhabditis elegans*, mitotic chromosome lengths become shorter

(*Ladouceur et al., 2015*; *Micheli et al., 1993*). The mechanistic basis of mitotic chromosome shape regulation remains largely speculative (*Heald and Gibeaux, 2018*).

Classical experiments on mitotic chromosomes indicated that DNA is organized into loops around a central protein scaffold (*Earnshaw and Laemmli, 1983*; *Paulson and Laemmli, 1977*). Two major chromosome-associated ATPases play pivotal roles in mitotic chromatid formation: DNA topoisomerase II (topo II) and the structural maintenance of chromosomes (SMC) family complex, condensin (*Kinoshita and Hirano, 2017*; *Kschonsak and Haering, 2015*). Data obtained with chromosome conformation capture assays are consistent with the model that mitotic chromosomes are arranged in a series of loops organized by condensins (*Naumova et al., 2013*; *Gibcus et al., 2018*; *Elbatsh et al., 2019*). Vertebrate cells express two forms of the condensin complex, condensin I and condensin II (*Ono et al., 2003*; *Hirano et al., 1997*). Condensin I is loaded onto chromatin exclusively during mitosis, whereas condensin II retains access to chromosomes throughout the cell cycle (*Hirota et al., 2004*; *Ono et al., 2004*; *Walther et al., 2018*). Condensin II plays a role in maintaining chromosome territories in *Drosophila* nuclei (*Rosin et al., 2018*; *Bauer et al., 2012*) and drives sister chromatid decatenation by topo II (*Nagasaka et al., 2016*). It has been proposed that condensin II acts first in prophase to anchor large outer DNA loops, which are further branched into shorter inner DNA loops by condensin I (*Gibcus et al., 2018*). This proposal is consistent with their localization as determined by super-resolution microscopy (*Walther et al., 2018*). In chicken DT40 cells, condensin II drives the helical positioning of loops around a centrally located axis, thus controlling the organization of long distance interactions (6–20 Mb), whereas condensin I appears to control shorter distance interactions (*Gibcus et al., 2018*). This organization of the condensin I and II loops is also consistent with their roles in maintaining lateral and axial compaction, respectively (*Green et al., 2012*; *Samejima et al., 2012*; *Bakhrebah et al., 2015*). In *Xenopus* egg extracts, in the presence of wildtype condensin I levels, condensin II depletion does not appear to change mitotic chromosome length, suggesting a reduced role for condensin II on these chromosomes (*Shintomi and Hirano, 2011*).

The prevailing model suggests that mitotic chromatin loops are formed by the dynamic loop extrusion activity of condensins (*Riggs, 1990*; *Nasmyth, 2001*; *Alipour and Marko, 2012*), although the molecular details of the process remain unclear (*Banigan and Mirny, 2020*; *Cutts and Vannini, 2020*; *Datta et al., 2020*). Single-molecule experiments using purified recombinant yeast and human condensin complexes demonstrated ATP-dependent motor activity and loop extrusion by yeast and human condensins (*Terakawa et al., 2017*; *Ganji et al., 2018*; *Kong et al., 2020*). Condensin-dependent loop extrusion in a more physiological *Xenopus* extract system has also been shown (*Golfier et al., 2020*). In silico experiments further suggest that a minimal combination of loop extruders (like condensin) and strand passage activity (such as topo II) can generate well-resolved rod-like sister chromatids from entangled, interphase-like DNA fibers (*Goloborodko et al., 2016a*). However, it remains unclear if loop extrusion can proceed on chromatin since condensins prefer to bind nucleosome-free DNA (*Kong et al., 2020*; *Zierhut et al., 2014*; *Shintomi et al., 2017*; *Toselli-Mollereau et al., 2016*; *Piazza et al., 2014*). Human and yeast condensin complexes are capable of loop extrusion through sparsely arranged nucleosomes in vitro (*Kong et al., 2020*; *Pradhan et al., 2021*), but mitotic chromatin adopts a more compact fiber structure (*Grigoryev et al., 2016*; *Ou et al., 2017*). Furthermore, large protein complexes such as RNA polymerases are able to limit loop extrusion by SMC protein complexes, such as bacterial condensins and eukaryotic cohesins (*Brandão et al., 2019*; *Hsieh et al., 2020*; *Krietenstein et al., 2020*). Nucleosomes also restrict the diffusion of cohesin in vitro (*Stigler et al., 2016*). Therefore, the effect of higher-order chromatin fiber structure and other mitotic chromatin proteins on processive loop extrusion by condensin remains unknown.

Interphase nuclei are segmented to chromosome territories, each of which contain highly entangled sister chromatids after replication (*Cremer and Cremer, 2010*; *Sundin and Varshavsky, 1981*; *Farcas et al., 2011*). Replicated pairs of chromatids are linked by cohesin during interphase, and cohesin removal during prophase (except at centromeres and other limited protected loci) promotes resolution of sister chromatids, together with actions of condensin II and topo II (*Nagasaka et al., 2016*; *Gandhi et al., 2006*). Different chromosomes are also largely unentangled in interphase HeLa cells (*Goundaroulis et al., 2020*; *Tavares-Cadete et al., 2020*), and Ki-67 localization on chromosome peripheries may act as a steric and electrostatic barrier to prevent interchromosomal entanglement during mitosis (*Cuylen et al., 2016*). However, some interchromosomal linkages were observed in metaphase (*Potapova et al., 2019*; *Marko, 2008*). Although it has been shown that active transcription

at rDNA results in mitotic interchromosomal links (*Potapova et al., 2019*), it remains unclear if mitotic chromosome compaction is coupled to resolution of these interchromosomal linkages.

One of the most abundant chromatin proteins beside core histones is the linker histone, which binds to the dyad of the nucleosome and tethers the two linker DNAs emanating from the nucleosome (*Bednar et al., 2017*; *Arimura et al., 2020*; *Zhou et al., 2015*; *Zhou et al., 2021*). In reconstitution experiments, linker histones cluster oligo-nucleosomes (*White et al., 2016*; *Li et al., 2016*) and promote liquid-liquid phase separation (*Gibson et al., 2019*; *Shakya et al., 2020*). In vivo, linker histones are also enriched in highly compact chromatin (*Izzo et al., 2013*; *Th'ng et al., 2005*; *Parseghian et al., 2001*). While core histones are evolutionarily highly conserved in eukaryotes, linker histones are much more diversified (*Izzo et al., 2008*). In the human and mouse genome, 11 H1 paralogs are found, among which some combination of six variants (H1.0–H1.5) is widely expressed in somatic cells (*Hergeth and Schneider, 2015*). In vertebrate oocytes and early embryos, H1.8 (also known as H1OO, H1foo, H1M, and B4) is the major linker histone variant (*Dworkin-Rastl et al., 1994*; *Wühr et al., 2014*). Immunodepletion of H1.8 from *Xenopus* egg extracts made mitotic chromosomes thinner and longer, causing defective chromosome segregation in anaphase (*Maresca et al., 2005*). However, the mechanism by which the linker histone affects large-scale chromosome length changes remains unknown.

Here we demonstrate that the linker histone H1.8 suppresses enrichment of condensins and topo II on mitotic chromosomes. In a reconstitution system with purified components, H1.8 inhibits binding of topo II and condensins to nucleosome arrays. Through a combination of chromosome morphological analysis and Hi-C, we show that H1.8 reduces chromosome length by limiting condensin I loading on chromosomes, while H1.8 limits chromosome individualization by antagonizing both condensins and topo II. This study establishes a mechanism by which the linker histone tunes the compaction and topology of mitotic chromosomes.

## Results

### Linker histone H1.8 limits enrichment of condensins and topo II on mitotic chromatin

Depletion of linker histone H1.8 in *Xenopus* egg extracts makes chromosomes thinner and elongated (see Figure 3C, *Maresca et al., 2005*). We asked if this phenotype may reflect the potential role of H1.8 in regulating condensins and TOP2A (the dominant topo II isoform in *Xenopus* egg extracts, *Wühr et al., 2014*), which are essential for mitotic chromosome compaction in *Xenopus* egg extracts (*Hirano and Mitchison, 1994*; *Adachi et al., 1991*; *Cuvier and Hirano, 2003*). In silico simulation analysis suggests that increasing the number of loop extruders (such as condensin I) on DNA, beyond a minimum threshold, makes chromosomes longer and thinner (*Goloborodko et al., 2016a*). Reducing condensin I levels on chromatin also made chromosomes shorter experimentally (*Shintomi and Hirano, 2011*; *Elbatsh et al., 2019*; *Fitz-James et al., 2020*). Although it has been reported that H1.8 depletion does not affect chromosomal enrichment of major chromatin proteins (*Maresca et al., 2005*), we therefore attempted to quantify chromatin-bound levels of condensins and TOP2A.

To investigate whether H1.8 regulates chromatin levels of condensins and topo II, we prepared mitotic chromosomes in *Xenopus laevis* egg extracts depleted of H1.8. Demembranated *X. laevis* sperm nuclei were added to either mock (ΔIgG) or H1.8-depleted (ΔH1) extracts from eggs arrested at meiotic metaphase II by cytostatic factor (CSF extracts; *Figure 1A and B*). Calcium was added to cycle the extract into interphase and induce functional nuclear formation, in which chromosomes were replicated. The corresponding depleted CSF extract was then added to generate metaphase chromosomes (*Shamu and Murray, 1992*). To eliminate the microtubule-dependent change in chromosome morphology, which may affect quantitative analyses of chromatin proteins, spindle assembly was inhibited using nocodazole. Chromosomes were fixed and the levels of TOP2A and condensin I subunit CAP-G were measured by immunofluorescence (*Figure 1C*). Depletion of H1.8 increased the levels of both CAP-G and TOP2A on mitotic chromosomes, while adding back recombinant H1.8 rescued the phenotype (*Figure 1D*). Identical results were obtained when immunofluorescence signal normalization was done by the minor groove DNA-binding dye Hoechst 33342 or by fluorescent dUTP that was incorporated during replication (*Figure 1—figure supplement 1A–C*). The dUTP quantitation also confirmed that H1.8 depletion did not affect DNA replication (*Figure 1—figure supplement 1B*),

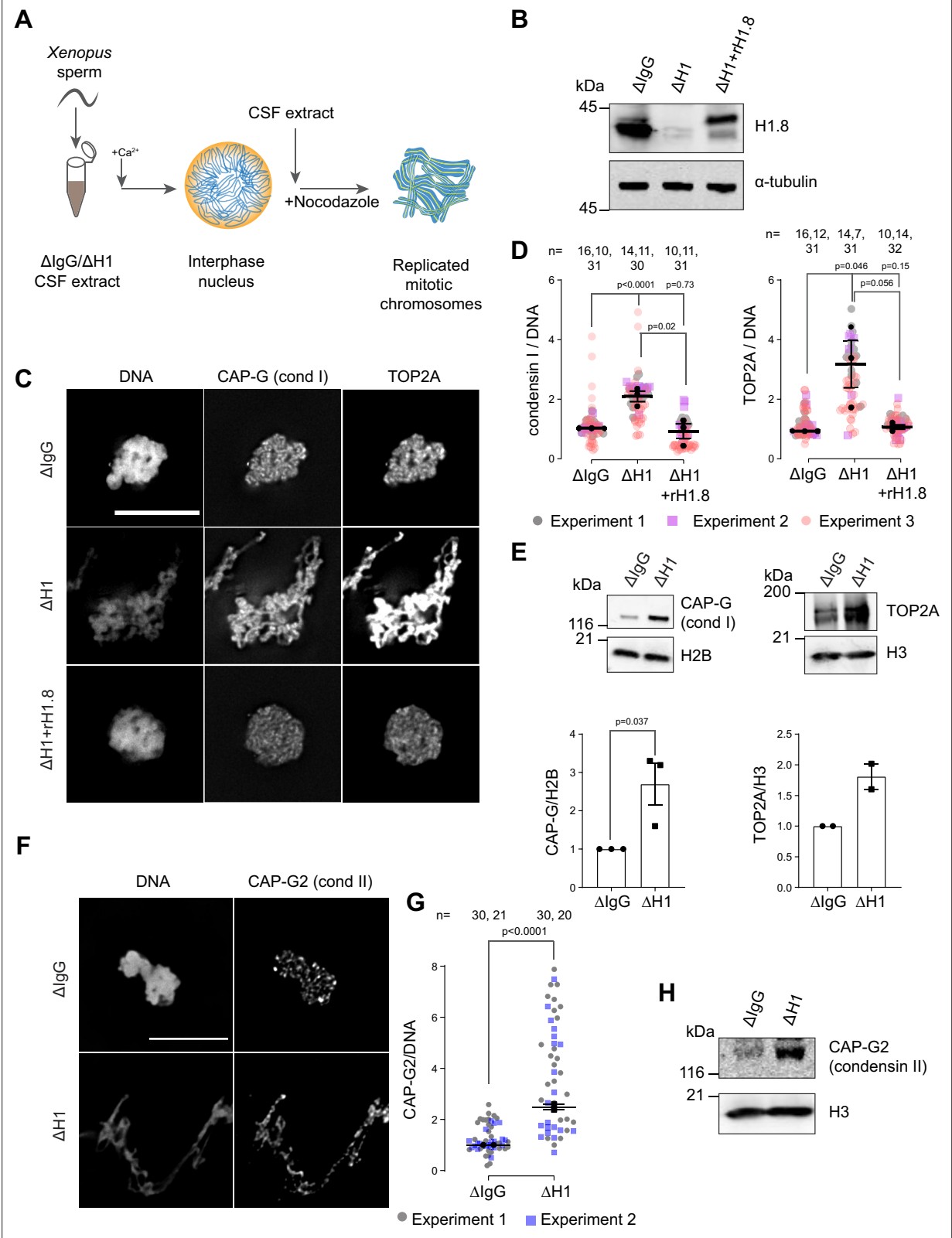

**Figure 1.** Linker histone H1.8 suppresses enrichment of condensins and TOP2A on mitotic chromatin. (**A**) Experimental scheme to generate replicated chromosomes in *Xenopus* egg extracts. (**B**) Western blots of total extracts showing depletion of H1.8 from *Xenopus* egg extracts and rescue with recombinant H1.8 (rH1.8). (**C**) Representative images of DNA (Hoechst 33342), CAP-G (condensin I), and TOP2A immunofluorescence on chromosomes in metaphase extracts treated with nocodazole in the indicated conditions. Chromosomes in each nucleus remain clustered in the presence of

*Figure 1 continued on next page*

*Figure 1 continued*

nocodazole. Bar, 10 µm. (**D**) Quantification of CAP-G (condensin I) and TOP2A immunofluorescence signals normalized to the DNA (Hoechst) signal for the indicated conditions. Each gray or magenta dot represents the average signal intensity of a single chromosome cluster (from one nucleus). Each black dot represents the median signal intensity from a single experiment. Bars represent mean and SEM of the medians of three independent experiments. (**E**) Western blots of mitotic chromatin purified from mock (ΔIgG) and H1.8-depleted (ΔH1) extracts (top) and quantification of band intensities normalized to H3 and H2B (below). Mean and SEM/range from three/two experiments respectively. (**F**) Representative images of CAP-G2 (condensin II) immunofluorescence on chromosomes in metaphase extracts with nocodazole in the indicated conditions. Bar, 10 µm. (**G**) Quantification of the CAP-G2 (condensin II) normalized to the DNA (Hoechst) signal for the indicated conditions. Each gray or purple dot represents the average signal intensity of a single chromosome cluster (from one nucleus). Each black dot represents the median signal intensity from a single experiment. Bars represent mean and range of the median of two independent experiments. (**H**) Western blots of mitotic chromatin purified from mock (ΔIgG) and H1.8-depleted (ΔH1) extracts. The *p*-values shown in (**D**) and (**E**) were calculated by an unpaired Student's *t*-test of the aggregate medians of three independent experiments, after confirming the statistical significance for each experimental dataset by a two-tailed Mann–Whitney *U*-test. The *p*-values shown in (**G**) were calculated on total data from two independent experiments using a two-tailed Mann–Whitney *U*-test. The number of nuclei imaged in (**D**) and (**G**) in each condition for each experiment is indicated in the figure.

The online version of this article includes the following figure supplement(s) for figure 1:

**Source data 1.** Source data for all the figures in *Figure 1* and its figure supplement.

**Source data 2.** Mass spectrometry data for chromatin purified from ΔIgG and ΔH1 metaphase sperm chromosomes.

**Figure supplement 1.** H1.8 depletion does not lead to global accumulation of DNA-binding proteins.

as reported previously (*Dasso et al., 1994*). The apparent increased signal intensities of CAP-G and TOP2A on chromatin in ΔH1 extracts were not the general consequence of elongated chromosome morphology as H1.8 depletion did not affect a panel of other chromatin proteins, regardless of their binding preference to nucleosomes or nucleosome-free DNA (*Zierhut et al., 2014*; *Figure 1—figure supplement 1D*). Enhanced chromosome binding of condensin I and TOP2A in ΔH1.8 extracts was biochemically confirmed by quantifying their levels on purified metaphase chromosomes by western blotting and by mass spectrometry (*Figure 1E*, *Figure 1—figure supplement 1E*). Mass spectrometry data also confirmed that H1.8 depletion did not result in a global enrichment of all chromatin-bound proteins. Condensin II levels on chromosomes also showed similar increases in ΔH1 extracts by immunofluorescence and western blots on purified chromosomes, although peptides of condensin II-specific subunits were not detected by mass spectrometry (*Figure 1F–H*, *Figure 1—figure supplement 1E*, *Figure 1—source data 2*).

## Linker histone H1.8 reduces binding of condensins and TOP2A to nucleosome arrays

Since linker histone depletion does not change the nucleosome spacing in *Xenopus* egg extracts (*Ohsumi et al., 1993*), we hypothesized that H1.8 depletion results in an increase in linker DNA that becomes accessible to condensins and TOP2A. To test this possibility, we reconstituted nucleosome arrays with purified histones and asked if H1.8 interferes with binding of recombinant human condensins and *X. laevis* TOP2A (*Kong et al., 2020*; *Ryu et al., 2010*; *Figure 2A*). The purity and the intact nature of the human condensin complexes was confirmed using mass photometry (*Figure 2—figure supplement 1*; *Verschueren, 1985*; *Young et al., 2018*; *Sonn-Segev et al., 2020*). As previously shown with human condensin I purified from HeLa cells (*Kimura et al., 2001*), our recombinant human condensin I, but not the ATP-binding defective Q-loop mutant (*Hassler et al., 2019*; *Hopfner et al., 2000*; *Löwe et al., 2001*; *Kong et al., 2020*), was able to rescue mitotic chromosome morphology defects caused by condensin I depletion in *Xenopus* extracts (*Figure 2—figure supplement 2A*), demonstrating that the recombinant human condensin I can also functionally replace *Xenopus* condensin I. The recombinant *X. laevis* TOP2A used was also able to rescue chromatid formation in ΔTOP2A extracts and was able to perform ATP-dependent kinetoplast decatenation in vitro (*Figure 2—figure supplement 2B and C*). The nucleosome array was composed of 19 tandem repeats of 147 bp Widom 601 nucleosome positioning sequence and 53 bp linker DNA (*Lowary and Widom, 1998*), where full occupancy of the array by a nucleosome core particle (NCP) and H1.8 to each repeat unit was confirmed by native polyacrylamide gel electrophoresis (PAGE) (*Figure 2—figure supplement 3A*).

Condensin I binds weakly to mononucleosomes with linker DNA (*Kong et al., 2020*). We observed similar weak binding of condensin I to nucleosome arrays, and this binding was stimulated by ATP (*Figure 2—figure supplement 3B*). Since this enhancement was not seen when magnesium

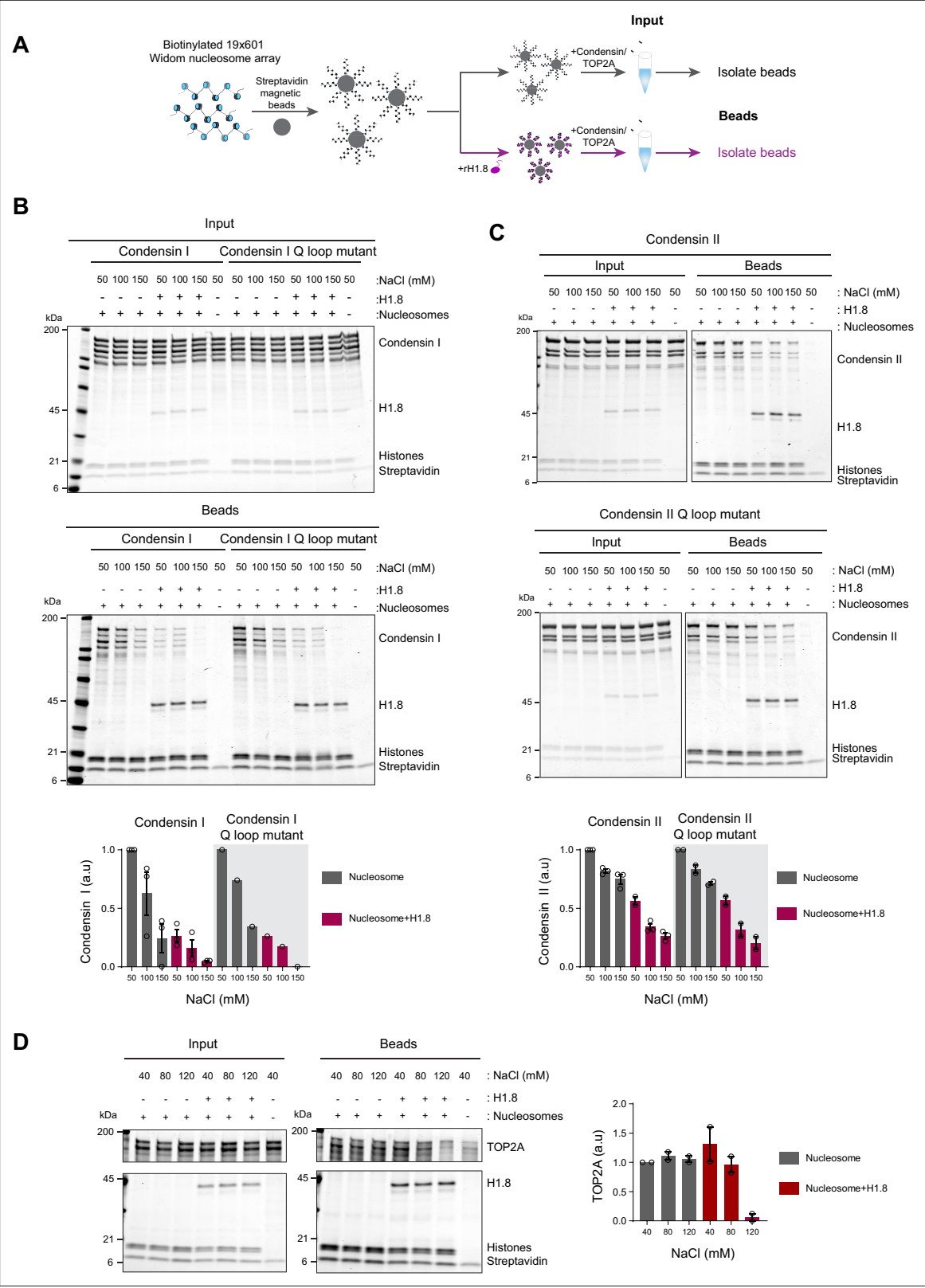

**Figure 2.** Linker histone inhibits binding of condensins and TOP2A to nucleosome arrays. (**A**) Experimental scheme for testing the effect of recombinant H1.8 (rH1.8) on binding of purified condensins and TOP2A to arrays of nucleosomes assembled on the Widom 601 nucleosome positioning sequence. (**B**) Coomassie staining of SDS-PAGE gels, showing input (top) and nucleosome array-bound fraction (middle) of condensin I, rH1.8, and core histones. The rightmost lanes represent the streptavidin beads-only negative control. Buffer contains 2.5 mM $MgCl_2$, 5 mM ATP, and indicated concentrations of

*Figure 2 continued on next page*

*Figure 2 continued*

NaCl. The band intensities of condensin I subunits were normalized to the histone bands and the binding at 50 mM NaCl for nucleosome arrays without H1.8. Mean and SEM of three independent experiments are shown (bottom). (**C**) Same as (**B**), except that nucleosome array binding of condensin II is shown. Mean and SEM (wildtype)/range (Q-loop mutant) of three (wildtype) or two (Q-loop mutant) independent experiments are shown. (**D**) Same as (**B**), except that nucleosome array binding of TOP2A in buffer containing 1 mM MgCl₂ is shown. Mean and range of two independent experiments are shown.

The online version of this article includes the following figure supplement(s) for figure 2:

**Source data 1.** Source data for all the figures in *Figure 2* and its figure supplements.

**Figure supplement 1.** Mass photometry of condensin complexes.

**Figure supplement 2.** Recombinant human condensin I and *X. laevis* TOP2A are functional.

**Figure supplement 3.** Condensin binding is inhibited by magnesium.

**Figure supplement 4.** H1.8 inhibits condensin binding to mononucleosomes.

concentration was higher than the ATP concentration (*Figure 2—figure supplement 3C*), we hypothesized that ATP might have enhanced condensin binding to the nucleosome arrays by chelating magnesium, which is known to induce chromatin compaction (*Finch and Klug, 1976*; *Eltsov et al., 2008*). Indeed, both EDTA and ATP, which chelate magnesium, increased binding of the Q-loop mutant of condensin I to the nucleosome array (*Figure 2—figure supplement 3D*), suggesting that condensin binding to the nucleosome array is sensitive to high magnesium concentration and not due to ATP binding of condensin I. In contrast, ATP did not stimulate condensin I binding to mononucleosomes (*Figure 2—figure supplement 3E*), indicating that excess magnesium may limit condensin binding through compaction of the nucleosome array.

In the buffer condition where excess ATP was present over magnesium, preloading of H1.8 to the nucleosome array reduced binding of both wild-type and the ATP-binding-deficient Q-loop mutant of condensin I at physiological salt concentrations (50–150 mM NaCl) (*Figure 2B*). H1.8 also suppressed binding of condensin II and its Q-loop mutant to the nucleosome array (*Figure 2C*), though, as expected (*Kong et al., 2020*), condensin II showed higher affinity to the nucleosome array than condensin I. We noticed that the subunit stoichiometry of condensin I and condensin II was altered in the bead fraction from that in the input. Since the condensin complexes were intact at the assay conditions (*Figure 2—figure supplement 1*), we suspect that non-SMC subunits of condensins were less stable on the nucleosome array than SMC subunits, while H1.8 reduced binding of all condensin subunits to the nucleosome array. The reduced binding of condensins in the presence of H1.8 can be explained by either direct competition between H1.8 and condensins for the linker DNA or by H1.8-mediated formation of a higher-order structure of the nucleosome array (*Song et al., 2014*; *Rudnizky et al., 2021*). The direct competition model can be tested by using mononucleosomes with linker DNAs instead of the nucleosome array since H1.8 binding to mononucleosomes does not promote higher-order structure or aggregation (*White et al., 2016*). Supporting the direct competition model, H1.8 reduced binding of condensin I to mononucleosomes (*Figure 2—figure supplement 4*).

We also examined if H1.8 interferes with binding of the recombinant *Xenopus* TOP2A to nucleosome arrays. As compared to condensins, TOP2A showed more stable binding to the nucleosome array, and H1.8 had no effect on TOP2A binding at low salt concentrations. However, H1.8 did reduce nucleosome array binding of recombinant *Xenopus* TOP2A at 120 mM NaCl (*Figure 2D*). Altogether, these data demonstrate that preloaded H1.8 on nucleosomes can directly interfere with binding of condensins and TOP2A to chromatin.

## Chromosome elongation by H1.8 depletion is due to increased chromatin-bound condensin I

If the increased amount of condensin I on chromatin is responsible for the chromosome elongation phenotype observed in ΔH1 extracts, reducing condensin I levels should reverse this phenotype. To measure lengths of mitotic chromosomes formed in *Xenopus* egg extracts, we assembled replicated metaphase chromosomes in extracts depleted with mock IgG, H1.8, CAP-G (condensin I), or CAP-D3 (condensin II) antibodies. These extracts were then diluted to disperse individualized chromosomes (*Figure 3A and B*; *Funabiki and Murray, 2000*). As reported previously (*Maresca et al., 2005*), average chromosome length increased by ~50% upon H1.8 depletion (*Figure 3C and D*). Supporting

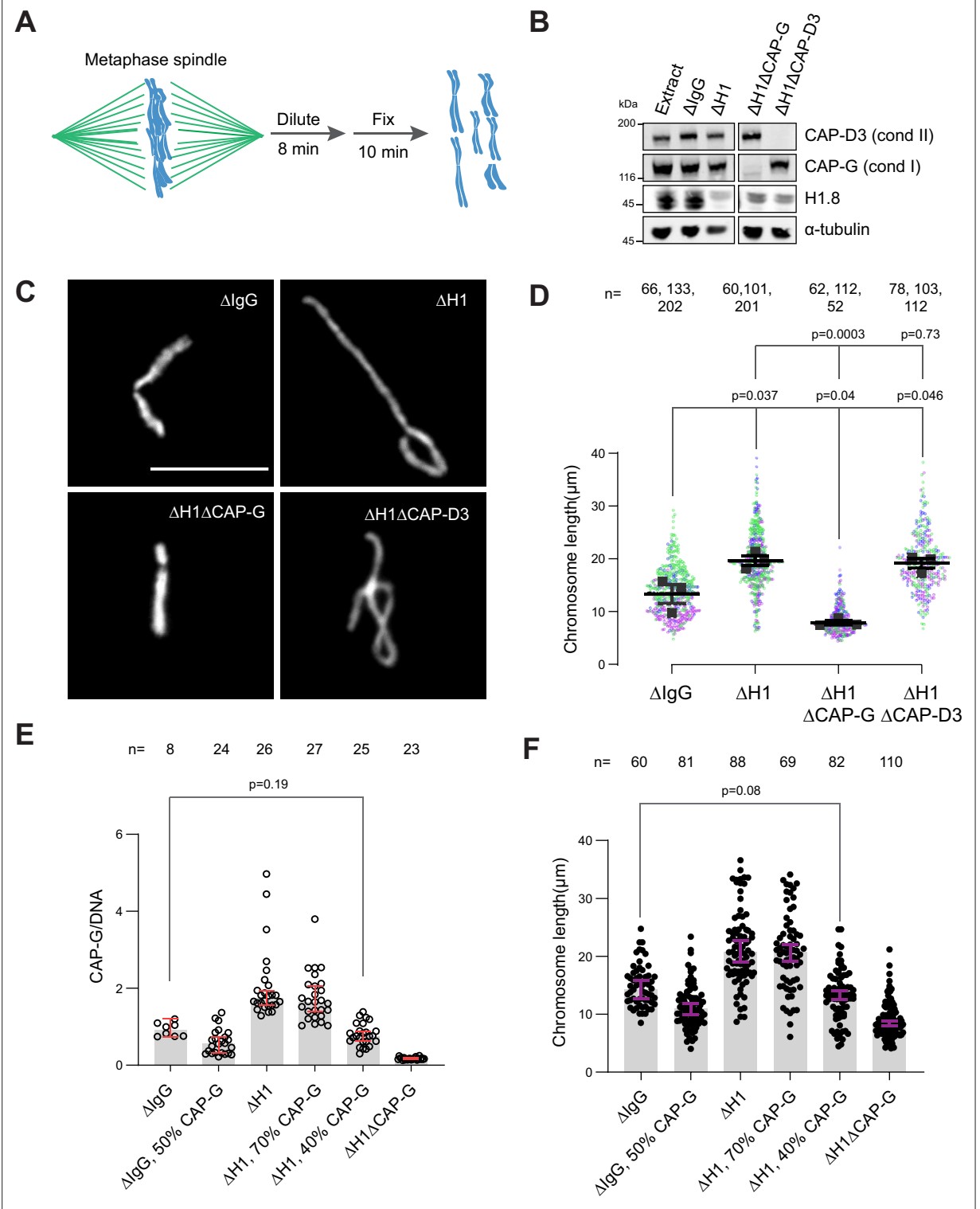

**Figure 3.** Chromosome elongation by H1.8 depletion is due to enhanced condensin I loading on chromatin. (**A**) Schematic of extract dilution to disperse individualized chromosomes. (**B**) Western blots of total egg extracts showing depletions of indicated proteins. (**C**) Representative images of mitotic chromosomes after dilution of indicated extracts. Bar, 10 μm. (**D**) Quantification of the chromosome length. Data distribution of the length of individual chromosomes from three independent experiments (green, purple, gray) is shown. Each black dot represents the median length of chromosomes from a single experiment. Bar represents mean and SEM of three independent experiments. (**E**) Quantification of CAP-G levels normalized to DNA signal (Cy3-dUTP) in the indicated conditions by immunofluorescence. Each dot represents the mean of CAP-G intensity normalized

*Figure 3 continued on next page*

Figure 3 continued

to DNA intensity of a single chromosome cluster (from one nucleus). The data plotted is median ± 95% CI. (**F**) Chromosome lengths in the indicated condition. Each dot represents length of a single chromosome. Bars represent median ± 95% CI. The p-values in (**D**) compare the median chromosome lengths in each condition and were calculated using an unpaired Student's *t*-test, and those in (**E**, **F**) compare the median values in a single experiment and were calculated using a two-tailed Mann–Whitney *U*-test. The number of nuclei (**E**) or chromosomes (**D, F**) imaged in each condition for each experiment is indicated above the figure.

The online version of this article includes the following figure supplement(s) for figure 3:

**Source data 1.** Source data for all the figures in *Figure 3* and its figure supplements.

**Figure supplement 1.** Condensin I loading determines chromosome length.

**Figure supplement 2.** TOP2A overloading is not responsible for chromosome elongation.

our hypothesis, when condensin I was co-depleted (ΔH1ΔCAP-G), chromosomes became even shorter than chromosomes in mock-depleted extracts (ΔIgG) (*Figure 3C and D*). In contrast, condensin II co-depletion (ΔH1ΔCAP-D3) did not change the chromosome length (*Figure 3D*). Chromosome length in condensin II-depleted extracts (ΔCAP-D3) was also indistinguishable from mock-depleted chromosomes (*Figure 3—figure supplement 1A*), consistent with the negligible effect of condensin II depletion in the presence of normal levels of condensin I in *Xenopus* egg extracts (*Shintomi and Hirano, 2011*).

If H1.8 regulates chromosome length by a mechanism independent of its role in condensin I inhibition, we expect that the chromosome lengths in ΔCAP-G extracts and ΔH1ΔCAP-G extracts would be different. Unfortunately, since severe defects in chromosome individualization in ΔCAP-G extracts prevented us from measuring chromosome lengths, this comparison was not possible (see Figure 6B). To circumvent this issue, we asked if similar condensin I levels on both ΔIgG and ΔH1 chromatin would result in similar chromosome lengths. Since condensin I binding to chromosomes is suppressed by H1.8, to load condensin I to chromosomes in ΔH1.8 extracts at the levels seen in control (ΔIgG) extracts, we assembled chromosomes in extracts where condensin I was partially depleted (*Figure 3—figure supplement 1B*). When mitotic chromosomes were assembled in ΔH1.8 extracts that contain 40% (of ΔIgG) condensin I (CAP-G), condensin I level on these chromosomes was equivalent to that on chromosomes in control (ΔIgG) extracts (*Figure 3E*, *Figure 3—figure supplement 1C*). Consistent with the hypothesis that H1.8 does not contribute to chromosome length regulation independently of condensin I, average chromosome length was essentially identical between these two conditions (ΔH1 40% CAP-G and ΔIgG; *Figure 3F*, *Figure 3—figure supplement 1D*).

As H1.8 depletion also results in the accumulation of TOP2A (*Figure 1*), we investigated whether the increased TOP2A also plays a role in the chromosome elongation in ΔH1 extracts. As TOP2A activity is essential for sperm decondensation (*Adachi et al., 1991*; *Shintomi et al., 2015*), we examined the chromosome length upon partial depletion of TOP2A (*Figure 3—figure supplement 2A*). Similar to previous reports from other systems (*Farr et al., 2014*; *Samejima et al., 2012*; *Nielsen et al., 2020*), mitotic chromosomes prepared in extracts with reduced TOP2A activity show slightly increased chromosome length (*Figure 3—figure supplement 2B*). Consistent with these observations, partial depletion of TOP2A in H1.8-depleted extracts also led to a further increase and not decrease in chromosome length (*Figure 3—figure supplement 2C–E*), suggesting that chromosome elongation in H1.8-depleted extracts was not caused by the increased level TOP2A on chromatin. In summary, these results demonstrate that H1.8 controls mitotic chromosome lengths primarily through limiting the chromosome binding of condensin I.

## H1.8 regulates condensin I-driven mitotic loop layer organization

Experimental and simulation studies have shown that mitotic chromosome length and width are sensitive to the number of condensin I molecules on a chromosome, through affecting the loop size (*Gibcus et al., 2018*; *Goloborodko et al., 2016a*; *Goloborodko et al., 2016b*; *Fitz-James et al., 2020*). Low levels of condensin I are predicted to result in fewer and larger loops, while higher levels of condensin I will lead to a larger number of smaller loops. Therefore, if linker histone H1.8 decreases chromosome length through limiting condensin I association with chromatin, we expect that the average size of mitotic condensin loops decreases upon H1.8 depletion. As an alternative way to quantitatively assess

the effect of H1.8 and condensins on mitotic chromosome organization, we used the chromosome conformation capture assay Hi-C (*Lieberman-Aiden et al., 2009*).

Hi-C contact probability maps were generated from replicated metaphase *X. laevis* sperm chromosomes 60 min after cycling back into mitosis in the presence of nocodazole. As seen in mitotic chromosomes in somatic cells (*Naumova et al., 2013*; *Gibcus et al., 2018*) and in early stages of mouse development (*Du et al., 2017*), all the Hi-C contact maps showed no checkerboard pattern commonly associated with interphase chromosome compartments and lacked any sign of topologically associating domains (TADs) (*Szabo et al., 2019*; *Figure 4A*; data available at GEO under accession no. GSE164434). Hi-C interaction maps are characterized by the decay in contact probability, *P*, as a function of the genomic distance, *s*. To derive quantitative information about the polymer structure of the mitotic chromosomes, we plotted the genome-wide average *P*(*s*) (*Figure 4B*). *P*(*s*) plots were consistent among two biological replicates (*Figure 4—figure supplement 1A*) and among different chromosomes (*Figure 4—figure supplement 1B*). The interaction decay profile of the control (ΔIgG) chromosomes was also qualitatively similar to mitotic chromosomes in both DT40 and human cells (*Naumova et al., 2013*; *Gibcus et al., 2018*; *Elbatsh et al., 2019*). Condensin I depletion (ΔCAP-G) caused a major change in the Hi-C map (*Figure 4A*), reflecting its severe morphological defects in mitotic chromosomes in *Xenopus* egg extracts (*Hirano et al., 1997*). Unlike in DT40 cells, where depletion of condensin I and condensin II reduces interactions at shorter (<6 Mb) and longer (>6 Mb) distances, respectively (*Gibcus et al., 2018*), condensin I depletion affected interactions at longer distances (~10 Mb), whereas condensin II depletion (ΔCAP-D3) did not cause recognizable changes in interactions at both long and short distances (*Figure 4—figure supplement 1C*). A second diagonal band, which is indicative of a strong helical organization in the chromosome axis (*Gibcus et al., 2018*), was not seen in *Xenopus* egg extracts (*Figure 4A*, *Figure 4—figure supplement 1C*), likely reflecting the minor contribution of condensin II in this system or perhaps due to a prolonged arrest in mitosis (*Gibcus et al., 2018*). In H1.8-depleted extracts (ΔH1), a dramatic decrease in interactions at long genomic distances (1–10 Mb) was observed (*Figure 4A and B*), as expected from the thinner chromosomes (*Figure 3*).

*P*(*s*) plots are useful to determine the underlying polymer structure of the chromosome such as average loop size (*Gibcus et al., 2018*). Specifically, the first derivative (slope) of the *P*(*s*) plots can reveal both the average loop size and the amount of DNA per layer of the rod-shaped mitotic chromosome (layer size) (*Gassler et al., 2017*; *Abramo et al., 2019*; *Gibcus et al., 2018*; *Figure 4C*). Average loop size can be estimated from the peak value in the derivative plot (*Gassler et al., 2017*; *Patel et al., 2019*). As the derivative plot in our data showed a peak in the 10 kb to 1 Mb range, we estimated that this peak location reflects the average loop size. Unlike mitotic chromosomes in DT40 or HeLa cell lines (*Gibcus et al., 2018*; *Abramo et al., 2019*), metaphase chromosomes from *Xenopus* extracts had a flattened peak (*Figure 4—figure supplement 1D*), possibly due to a larger variation in loop sizes. Although it is difficult to estimate the exact loop size from these plots, chromosomes from ΔH1 extracts showed a reproducible shift of the peak towards smaller genomic distances (*Figure 4—figure supplement 1D*). This is consistent with a decrease in loop size due to increased condensin I accumulation on chromatin upon H1.8 depletion. Derivative plots generated from Hi-C maps of dispersed chromosomes in diluted extracts (*Figure 3A*) showed a similar shift in the peak towards smaller genomic distances upon H1.8 depletion (*Figure 4C*). The derivative plots from these dispersed chromosomes showed a better-defined peak, allowing a coarse loop size estimate of ~140 kb in control (ΔIgG) extracts and ~110 kb in ΔH1 extracts using the previously reported derivative peak heuristic (*Gassler et al., 2017*). These coarse loop size estimates are comparable to the estimated size of condensin I-driven loops in DT40 and HeLa cells (*Gibcus et al., 2018*; *Naumova et al., 2013*).

*P*(*s*) plots for mitotic chromosomes display three regimes that are typical for relatively stiff rod-shaped conformation (*Naumova et al., 2013*; *Gibcus et al., 2018*): for small genomic distance, the contact probability is dominated by interactions between pairs of loci located within loops (up to 100–200 kb). For loci separated by up to a few megabases, the contact probability decays slowly with genomic distance. These interactions mostly reflect contacts between loci located in different loops that are relatively closely packed as a radial layer of loops around the central axis (intralayer regime). The third regime is characterized by a steep decay in contact probability at several megabases. This represents pairs of loci separated by a relatively large distance along the axis of the rod-shaped

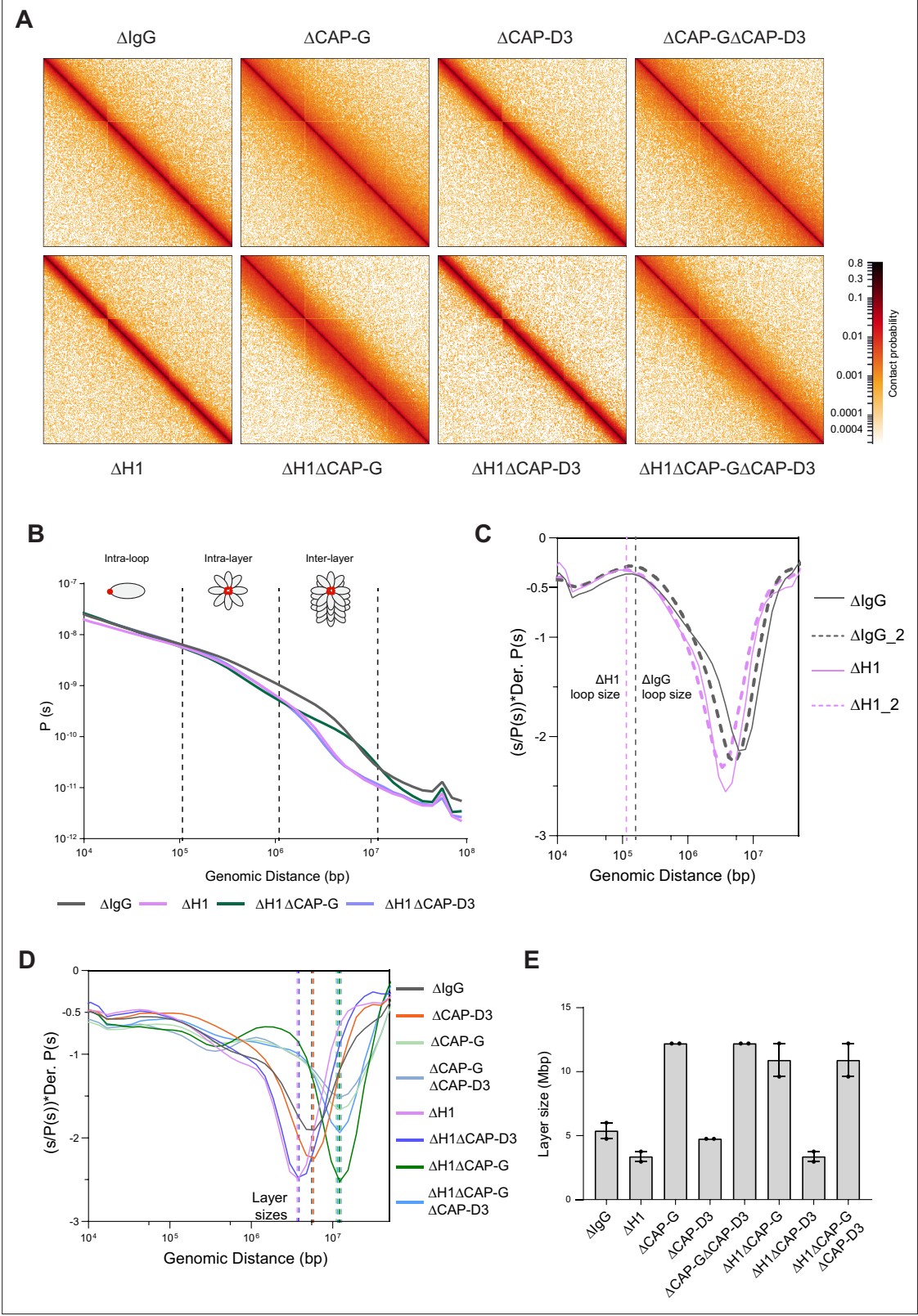

**Figure 4.** Effects of H1 and/or condensin I and II depletion on mitotic genome folding. (**A**) Hi-C maps of metaphase *X. laevis* chromosome 3S , binned to 250 kb, in the indicated condition. (**B**) Genome-wide average contact probability decay curves for the indicated conditions showing the changes in longer distance interactions. (**C**) Derivative plots of the average contact probability decay curves for dispersed chromosomes from mock (ΔIgG) and H1.8-depleted extracts (ΔH1) showing the change in estimated loop size. The solid and dotted lines are from two independent biological replicates.

*Figure 4 continued on next page*

*Figure 4 continued*

(**D**) Derivative plots of the genome-wide average contact probability decay curves in the indicated conditions. The dotted lines indicate the layer sizes for each plotted condition. (**E**) Estimates of layer size from derivatives of genome-wide probability decay curves upon depletion of H1.8 and CAP-G (condensin I) or CAP-D3 (condensin II). The mean and range of two biological replicates are shown.

The online version of this article includes the following figure supplement(s) for figure 4:

**Source data 1.** Source data for all the figures in *Figure 4* and its figure supplements.

**Figure supplement 1.** Effects of H1, condensin I, and condensin II depletion on mitotic genome folding.

**Figure supplement 2.** H3-H4 depletion leads to even smaller loop sizes.

chromosome so that they very rarely interact (interlayer regime). The size of these layers can be obtained from the derivative of $P(s)$, where the steep drop in contact probability curve is marked by sharp drop in the derivative value. For control chromosomes, the average amount of DNA in each layer of loops (layer size) was approximately 5 Mb, given the steep decay in contact probability observed for loci separated by more than 5 Mb for both replicates (*Figure 4B and D*). For H1.8-depleted chromosomes, we observed a smaller layer size of around 3.5 Mb (*Figure 4B and D*). This 1.5-fold reduction of DNA content of each layer would explain the observed 1.5-fold increase in chromosome length (*Figure 3D*). Further, we can derive the number of loops per layer by dividing the layer size by the loop size in the corresponding condition. In both control (ΔIgG) and H1.8-depleted extracts (ΔH1), the number of loops per layer was ~40. This indicates that the change in the layer size upon H1.8 depletion is a result of decreased loop size, while the number of loops per layer is not affected.

The layer size in H1.8/condensin I co-depleted extracts (ΔH1ΔCAP-G) was larger than those in control extracts (ΔIgG) (*Figure 4D and E*). Further supporting the idea that the layer size anticorrelates with the chromosome length, chromosomes in ΔH1ΔCAP-G extracts were indeed shorter than those in control ΔIgG extracts (*Figure 3D and F*). Condensin II co-depletion (ΔH1ΔCAP-D3) did not affect the layer size (3.5 Mb) and also the chromosome length (*Figures 4D, E and 3D*). Condensin II depletion alone (ΔCAP-D3) also did not affect the layer size, consistent with the observed lack of change in the chromosome length (*Figure 4D and E*, *Figure 3—figure supplement 1A*). Taken together, these data support the hypothesis that H1.8 limits the condensin I level on chromatin and shortens chromosome lengths, allowing each condensin I to form a longer loop, and consequentially tuning the amount of DNA present in each layer. The data also suggest that unlike in chicken DT 40 cells condensin I and not condensin II plays the dominant role in the organization of loop layers (*Gibcus et al., 2018*).

Since condensin I binding to DNA is also limited by nucleosomes (*Zierhut et al., 2014*; *Kong et al., 2020*; *Shintomi et al., 2017*), we expected that loss of nucleosomes would similarly reduce the loop and layer sizes. To test this, we depleted H3-H4 tetramers using an antibody to acetylated lysine 12 of histone H4 (H4K12ac; *Zierhut et al., 2014*) and generated metaphase chromosomes for Hi-C (*Figure 4—figure supplement 2A*). Since *X. laevis* sperm contains preloaded paternal H3-H4 (*Shechter et al., 2009*), the number of nucleosomes in our metaphase chromosomes was expected to be reduced by at most 50 % . As nucleosomes occupy a large majority of the genomic DNA (*Lee et al., 2007*; *Chereji et al., 2019*), even a partial histone depletion (ΔH3-H4) increased chromatin-bound condensin I beyond that of H1.8 depletion (ΔH1) (*Figure 4—figure supplement 2B*). Consequentially, the layer size in ΔH3-H4 extracts became much smaller (around 500 kb) than in ΔH1 extracts (*Figure 4—figure supplement 2C*). Assuming that the number of loops per layer is similar in these chromosomes (~40), the loop size estimate is 12 kb, which is much shorter than in H1.8 depletion. Altogether these results suggest that global occupancy of nucleosomes and linker histones can affect DNA loop size and chromosome through controlling the number of condensin molecules on the chromatin fiber.

## H1.8 suppresses condensin-driven mitotic chromosome individualization

Condensins and topo II act in concert to generate mitotic chromosomes from decondensed interphase nuclei (*Cuvier and Hirano, 2003*). Both experimental observations and in silico experiments suggest that condensin can drive decatenation of sister chromatids (*Goloborodko et al., 2016a*; *Nagasaka et al., 2016*; *Marko, 2009*) even during metaphase arrest (*Piskadlo et al., 2017*). In addition, it has

been suggested that condensin-mediated chromosome compaction also promotes chromosome individualization (*Brahmachari and Marko, 2019*; *Sun et al., 2018*), though it remains to be established if different linear chromosomes (non-sisters) are catenated with each other even after completion of mitotic compaction since Ki-67 on chromosome surfaces may act as a barrier to prevent interchromosomal DNA interaction during mitosis (*Cuylen et al., 2016*).

To assess the role of H1.8, condensins, and topo II in chromosome individualization, we used two different assays (*Figure 5A*). The first was to measure the three-dimensional surface area of metaphase chromosome clusters. In the presence of the microtubule depolymerizing drug nocodazole, metaphase chromosomes derived from each nucleus clustered to form tight ball-like structures (*Figure 5B*). This clustering can be quantified by the large reduction in the surface area of metaphase chromosome clusters over control nuclei (*Figure 5C*, *Figure 5—figure supplement 1A*). This clustering suggests that, in control extracts, chromosomes are predisposed to accumulate interchromosomal contacts (*Figure 5A*). To verify if these interchromosomal contacts are topological in nature, we physically dispersed the clustered chromosomes by extract dilution (*Figure 5A*; *Funabiki and Murray, 2000*). Since this dilution procedure involves mechanical dispersal by hydrodynamic forces, we expect that chromosomes would remain clustered after dilution only in the presence of strong unresolved interchromosomal contacts. Indeed, when ICRF-193, the drug that inhibits topo II-dependent catenation/decatenation without leaving double-strand DNA breaks (*Tanabe et al., 1991*), was added to egg extracts at the mitotic entry, chromosome individualization was completely blocked, forming large chromosome clusters after extract dilution (*Figure 5—figure supplement 1B–D*, ICRF-50 min). Even when ICRF-193 was added to metaphase egg extracts after completion of metaphase chromatid formation, but 2 min before extract dilution (*Figure 5—figure supplement 1B*, bottom), efficiency of chromosome individualization decreased (*Figure 5—figure supplement 1C and D*, ICRF-2 min). These results suggest that substantial interchromosomal topological catenations remain unresolved in metaphase and that these can be resolved by TOP2A during mechanical dispersion (via spindle or dilution). To quantify the clustering of unresolved chromosomes, we stained the coverslips for CENP-A, the centromere-specific histone H3 variant, which marks a centromere locus per chromosome (*Edwards and Murray, 2005*). While each fully individualized chromosome showed one CENP-A focus (CENP-A doublet, representing centromeres of a paired sister chromatids, is counted as one focus), multiple CENP-A foci were observed in a cluster of chromosomes. Since even fully individualized chromosomes stochastically interacted during chromosome dispersion, we redefined the chromosome cluster here for the chromosome mass containing four or more CENP-A foci (*Figure 5D and E*, *Figure 5—figure supplement 1E*). Although chromosomes from each nucleus tightly compact into a ball-like mass in nocodazole treated metaphase extracts (*Figure 5B*), they individualized normally by dispersion upon extract dilution (*Figure 5D and E*, *Figure 5—figure supplement 1E*). This suggests that the interchromosomal contacts accumulated in the absence of spindles can be resolved by mechanical dispersion. TOP2A inhibition by treatment with ICRF-193 however almost completely blocked chromosome individualization in nocodazole-treated extracts (*Figure 5D and E*, *Figure 5—figure supplement 1E*), confirming the accumulation of interchromosomal catenations in clustered chromosomes.

Since condensins promote sister chromatid decatenation through topo II (*Baxter et al., 2011*; *Goloborodko et al., 2016a*), we then asked if H1.8-mediated suppression of condensins limits chromosome individualization and resolution of interchromosomal linkages. If so, H1.8 depletion may reduce the minimum required level of condensin activities to support chromosome individualization. In metaphase mock-depleted (ΔIgG) and H1.8-depleted (ΔH1) extracts with replicated chromosomes, the extract dilution procedure (*Figure 3A*) resulted mostly in physically separated single chromosomes but also a small number of clumped chromosomes (*Figure 6B*, *Figure 6—figure supplement 1C*). To quantify chromosome individualization, we measured chromosome clustering using CENP-A foci (*Figure 6—figure supplement 1*), and also independently by chromosome morphology-based classification (*Figure 6—figure supplement 1C*). Condensin I depletion (ΔCAP-G) resulted in defective chromosome individualization, suggesting that condensin I activity drives resolution of interchromosomal entanglements, and that condensin II is not sufficient to resolve these interchromosomal links in this background. Strikingly, co-depletion of H1.8 and condensin I (ΔH1ΔCAP-G) effectively rescued chromosome individualization without detectable CAP-G on chromatin (*Figure 6A–D*, *Figure 6—figure supplement 1D*). This apparent bypass of condensin I requirement in chromosome individualization required condensin II as chromosome individualization failed in the triple-depleted extracts

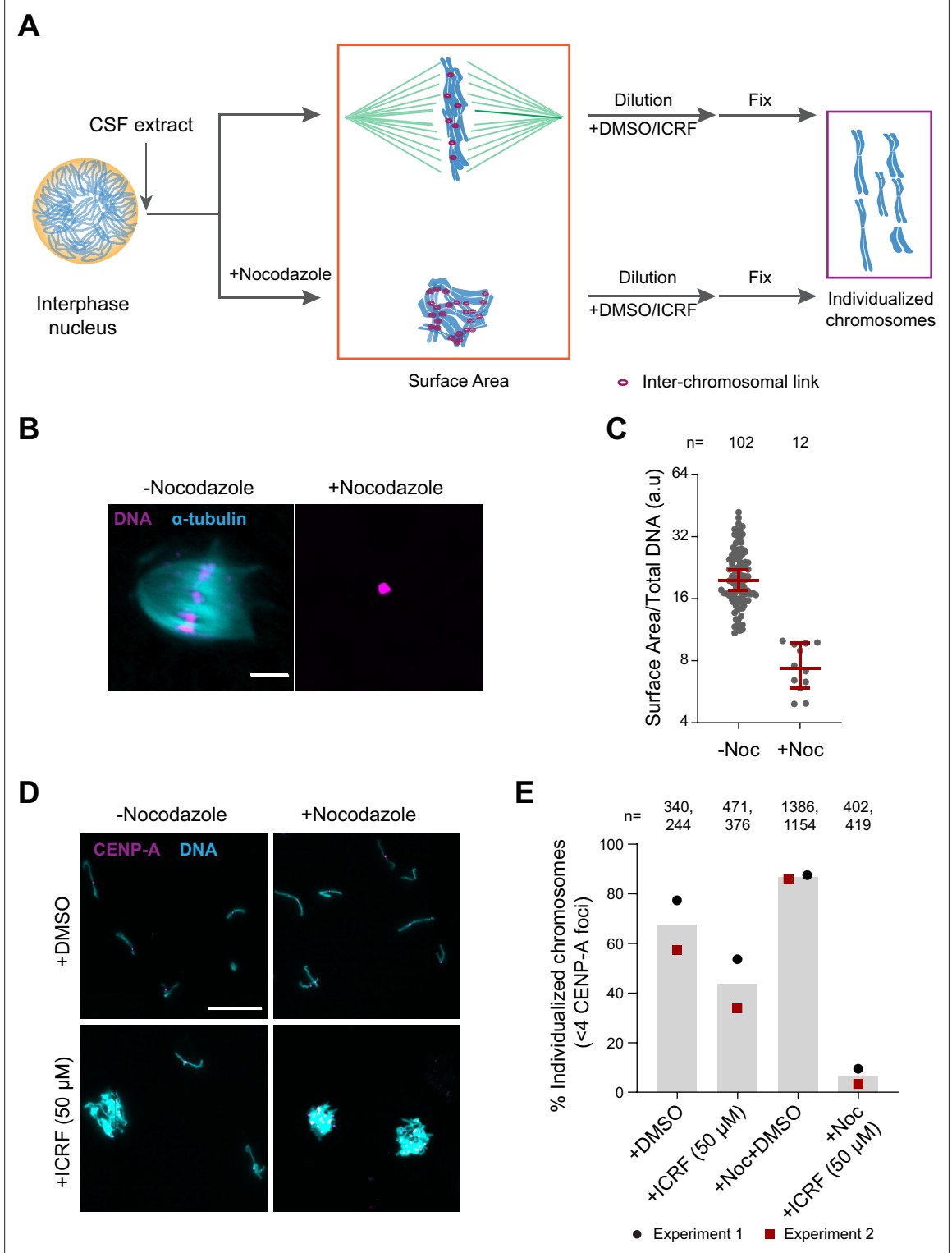

**Figure 5.** Substantial interchromosomal links remain in metaphase chromosomes. (**A**) Schematic showing the dispersal protocol and the different stages for the two chromosome individualization measurements. (**B**) Representative images of metaphase chromosomes in extracts containing Alexa Fluor 647-labeled tubulin (blue) with and without 10 μg/ml nocodazole. DNA was visualized using Hoechst 33342 (magenta). Scale bar is 20 μm. (**C**) Quantification of the three-dimensional surface area normalized to the total DNA intensity in (**B**). Each dot represents a single chromosomal mass. The number of masses quantified in each condition is indicated. (**D**) Representative images of the dispersed chromosomal masses in the indicated conditions. Scale bar is 20 μm. (**E**) Percent frequency of individualized chromosomes (chromosomes in DNA masses with <4 CENP-A foci) in the

*Figure 5 continued on next page*

*Figure 5 continued*

indicated conditions. Data from two biological replicates is shown. The number of masses quantified in each condition is indicated.

The online version of this article includes the following figure supplement(s) for figure 5:

**Source data 1.** Source data for *Figure 5* and its figure supplement.

**Figure supplement 1.** Regulation of chromosome individualization by topo II, condensins, and H1.8 in *Xenopus* egg extracts.

(ΔH1ΔCAP-GΔCAP-D3; *Figure 6B,C*). The increased condensin II in ΔH1ΔCAP-G extracts may replace the function of condensin I by condensin II in the absence of H1.8 (*Figure 6E*). These results demonstrate that H1.8-mediated suppression of condensin enrichment on chromatin limits chromosome individualization during mitosis.

## H1.8 prevents chromosomes from hyper-individualization by suppressing condensins and topo II

Since TOP2A-mediated decatenation played a role in resolving interchromosomal links, we next examined the functional significance of H1.8-mediated suppression of TOP2A enrichment on mitotic chromosomes (*Figure 1C–E*), asking if H1.8 depletion could reduce the required TOP2A level in extracts for chromosome individualization. Complete loss of TOP2A inhibits decompaction of sperm nuclei, a process associated with replacement of protamines with histones (*Adachi et al., 1991*), so we addressed this question using extracts partially depleted of TOP2A (*Figure 7A*). We first generated nuclei with replicated chromosomes in extracts containing the normal level of TOP2A, and then the extracts were diluted with ΔTOP2A extracts to reduce the total TOP2A level to 25% (*Figure 7—figure supplement 1A*). Under this condition, the level of chromosome-associated TOP2A also reduced to 25% (*Figure 7B*). Upon reduction of TOP2A to 25%, the frequency of unindividualized chromosome clusters increased about threefold in ΔIgG background (*Figure 7C*). However, in the absence of H1.8, extracts with 25% levels of TOP2A were still able to support maximum level of chromosome individualization as chromosome-associated levels of TOP2A became equivalent to untreated control extracts (*Figure 7C*). These data suggest that chromosome individualization is sensitive to TOP2A levels on chromatin, and that H1.8 suppression of TOP2A plays a role in suppressing chromosome individualization.

These data suggest that increased condensins and topo II levels on chromosomes in H1.8-depleted extracts reduce the number of links between chromosomes. To assess the consequence of these reduced interchromosomal links, we measured the chromosome clustering in nocodazole-treated extracts. As shown earlier, chromosomes of each nucleus in nocodazole-treated control extracts cluster into tight balls (*Figure 5B,C*). In contrast, individual chromosomes were more readily distinguished by DNA staining and spread to larger area in ΔH1 extracts, suggesting that the H1.8-mediated suppression of chromosome individualization is responsible for the chromosome clustering when spindle assembly is compromised (*Figure 7D,E*). Although condensin I depletion (ΔCAP-G) led to failed chromosome individualization (*Figure 6B,C*), these chromosomes do not show increased clustering over those seen in control (ΔIgG) extracts. This is consistent with the notion that chromosomes may already be maximally clustered in control extracts. However, H1.8 condensin I co-depleted extracts (ΔH1ΔCAP-G) showed reduced clustering compared to ΔIgG extracts, despite the shorter average chromosome length in ΔH1ΔCAP-G extracts (*Figure 3D,F*), suggesting that chromosome spreading upon H1.8 depletion is not primarily driven by chromosome elongation but by chromosome hyper-individualization. Since TOP2A is enriched on chromosomes upon H1.8 depletion in a condensin-independent manner (*Figure 7F*, *Figure 7—figure supplement 1B*), we then asked if this increased TOP2A plays a role in this increased chromosome spreading. Partial TOP2A depletion had no effect on clustering in control (ΔIgG) extracts since these chromosomes were already tightly packed together (*Figure 7G,H*–25% TOP2A). TOP2A depletion in H1.8-depleted extracts also did not reduce the chromosome spreading even though TOP2A levels on these chromosomes became comparable to that of the control (ΔIgG-100%; *Figure 7G,H*), perhaps because increased condensin loading can compensate for reduced TOP2A activity. However, partial TOP2A depletion reduced chromosome spreading in H1.8/condensin I co-depleted (ΔH1ΔCAP-G) extracts, suggesting that suppression of both condensin I and topo II by H1.8 keeps egg extract chromosomes together during mitosis even in the absence of spindle microtubules.

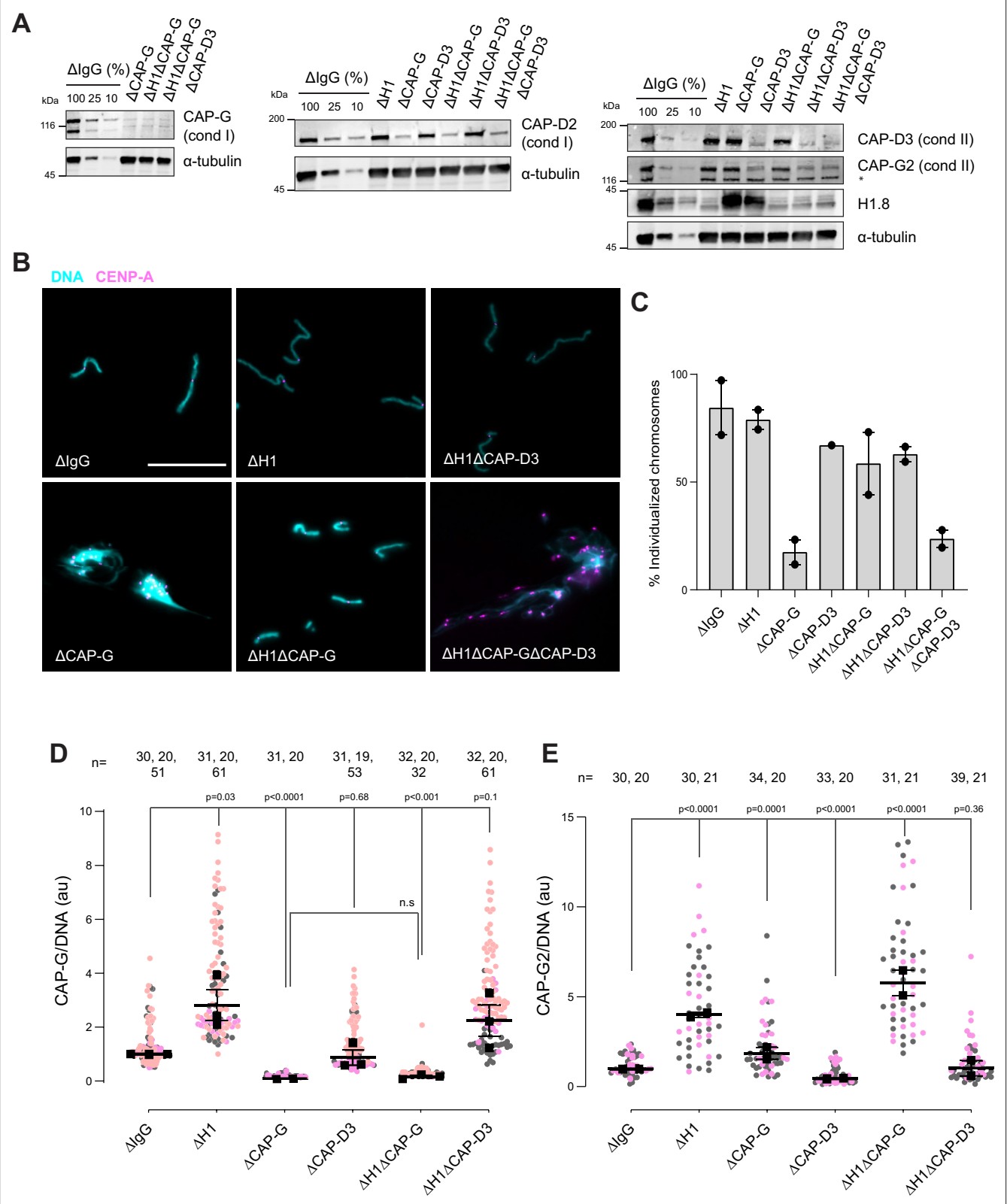

**Figure 6.** H1.8 suppresses condensin to limit chromosome individualization. (**A**) Western blots of total egg extracts showing depletion levels in extract of condensin I and condensin II using the CAP-G and CAP-D3 antibodies, respectively. * represents non-specific band. (**B**) Representative images of chromosomes after extract dilution, which disperses individualized chromosomes. DNA and centromere-associated CENP-A immunofluorescence are shown. Bar, 20 μm. (**C**) Percent frequency of individualized chromosomes (chromosomes in DNA masses with <4 CENP-A foci) in the indicated

*Figure 6 continued on next page*

*Figure 6 continued*

conditions. A large majority of DNA masses with no CENP-A foci are derived from ΔCAP-D3 extracts, where CENP-A loading is compromised (**Bernad et al., 2011**). DNA masses and CENP-A foci were identified using Otsu's thresholding algorithm and CENP-A foci in a binarized DNA mask were counted. The numbers of DNA masses counted in each condition were as follows: ΔIgG (502, 643), ΔH1 (1279, 839), ΔCAP-G (447, 170), ΔCAP-D3 (937), ΔH1ΔCAP-G (1565, 301), ΔH1ΔCAP-D3 (1536, 300), ΔH1ΔCAP-GΔCAP-D3 (300, 156). (**D**) Quantification of CAP-G (condensin I) immunofluorescence normalized to the DNA signal for the indicated conditions. Each gray or orange dot represents the average signal intensity of a single chromosome cluster (from one nucleus). Each black dot represents the median signal intensity from a single experiment. Bars represent mean and range of the medians of two independent experiments. (**E**) Quantification of CAP-G2 (condensin II) immunofluorescence intensity, normalized to the DNA signal for the indicated conditions. Each gray or magenta dot represents the average signal intensity of a single chromosome cluster (from one nucleus). Each black dot represents the median signal intensity from a single experiment. Bars represent mean and range of the medians of two independent experiments. The *p*-values in (**D**) and (**E**) were calculated by an unpaired Student's *t*-test and a two-tailed Mann–Whitney *U*-test respectively. The number of nuclei imaged in each condition in (**D**) and (**E**) in each experiment is indicated above the figures.

The online version of this article includes the following figure supplement(s) for figure 6:

**Source data 1.** Source data for all the figures in **Figure 6** and its figure supplement.

**Figure supplement 1.** Additional data for chromosome clustering.

## Discussion

It has been thought that the linker histone H1 promotes local chromatin compaction through stabilizing linker DNA and facilitating nucleosome-nucleosome interaction (**Song et al., 2014**; **White et al., 2016**; **Li et al., 2016**). Here we demonstrated that H1.8 also plays a major role in regulation of long-range DNA interaction through controlling chromatin loading of condensins. Although condensins and topo II are activated in mitosis to drive chromosome segregation (**Kimura et al., 1998**; **Hirano and Mitchison, 1991**), we showed that their functionalities are antagonized by H1.8 to tune chromosome length and prevent hyper-individualization, which increases surface area (**Figure 8**, **Table 1**). Thus, the linker histone H1, which can promote local chromatin compaction through promoting nucleosome-nucleosome interaction, can also suppress chromosome individualization and long-range DNA folding.

Nucleosomes reduce binding of condensins to DNA in vitro (**Kong et al., 2020**), in vivo (**Toselli-Mollereau et al., 2016**; **Piazza et al., 2014**), and in *Xenopus* egg extracts (**Zierhut et al., 2014**; **Shintomi et al., 2017**). Now we showed that the linker histone H1.8 limits binding of condensin I and II to chromatin both in vitro and in *Xenopus* egg extracts. H1.8 suppressed condensin binding on both mononucleosomes and nucleosome arrays, suggesting that H1.8 is able to compete out condensins for the same linker DNA targets (**Rudnizky et al., 2021**), though the capacity of linker histones to promote higher-order structures or phase separation may also limit the access of condensin (**Song et al., 2014**; **Gibson et al., 2019**). Since condensin I subunits are most abundant chromatin proteins whose levels were enhanced by H1.8 depletion (**Figure 1—figure supplement 1E**, **Figure 1—source data 2**), and chromosome lengths can be dictated by the amount of condensins in a manner independently of H1.8 (**Figure 3F**), we propose that regulating linker histone stoichiometry could serve as a rheostat to control chromosome length through tuning the condensin level on chromatin (**Figure 8**). Although such a linker histone-mediated chromosome length shortening seems odd in the large oocyte cells, mitotic chromosome length is constrained not only by the cell size but also by the spindle size (**Schubert and Oud, 1997**). As spindle size in *Xenopus* embryos does not scale with the cell size during the early divisions (**Wühr et al., 2008**), lack of chromosome clearance from the spindle midzone due to elongated anaphase chromosomes may result in chromosome breakage by the cytokinesis (**Maresca et al., 2005**; **Janssen et al., 2011**). By keeping the spindle and chromosome length short, variations in duration for chromosome segregation would be reduced to better synchronize cell division. Since the larger cell size correlates with reduced mitotic checkpoint strength (**Galli and Morgan, 2016**; **Minshull et al., 1994**), limiting chromosome length might be important for their timely and synchronous segregation during the rapid early embryonic cell divisions.

Since the average loop size is an aggregate result of loop extrusion rate, processivity, and number of loop extruders (**Goloborodko et al., 2016b**; **Alipour and Marko, 2012**), it is unclear from our data whether H1.8 affects the loop extrusion kinetics of a single condensin molecule. However, our observation is consistent with the in silico simulation showing that increasing the number of loop extruders makes chromosomes thinner and longer by reducing average loop size (**Goloborodko et al., 2016a**;

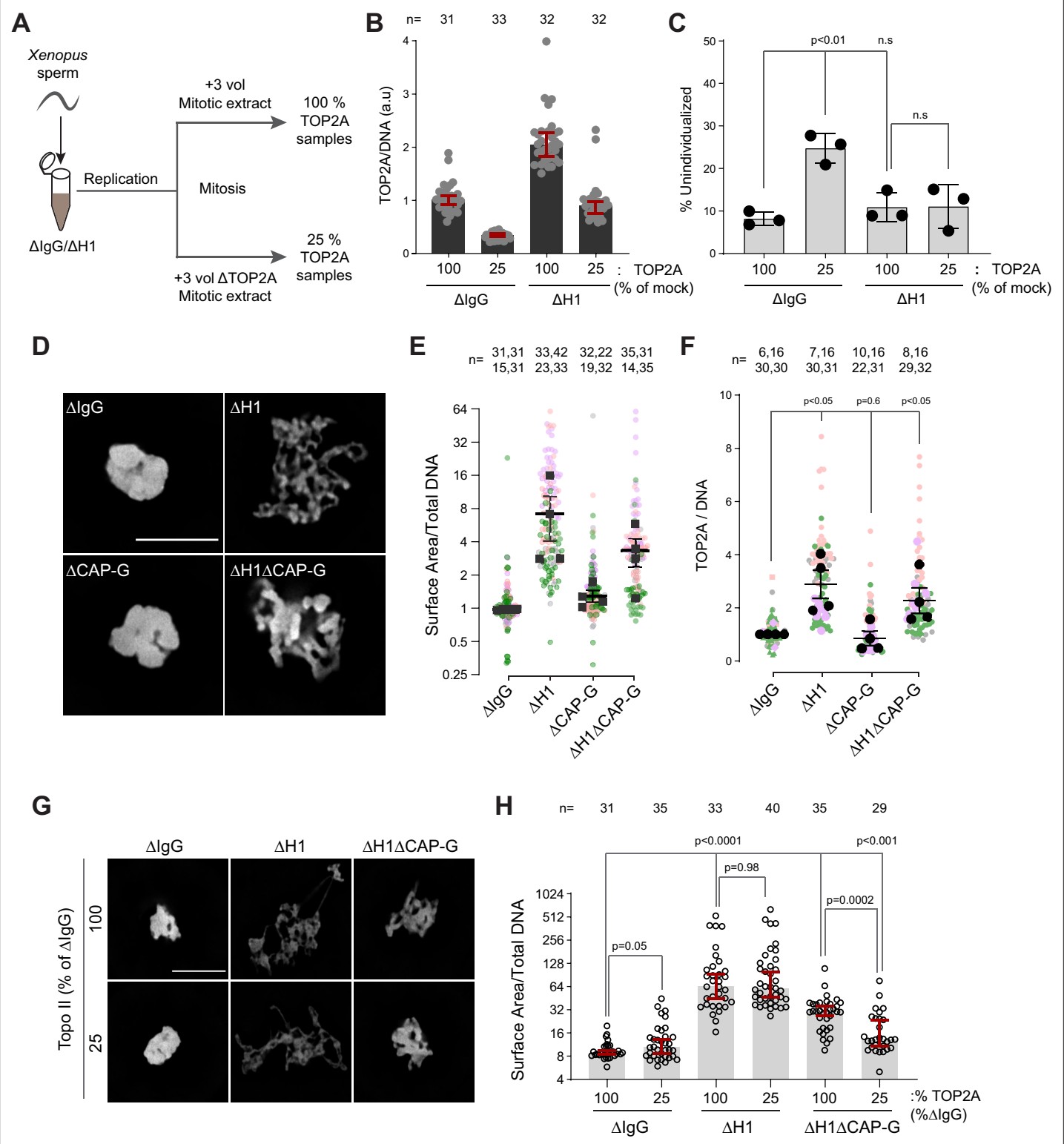

**Figure 7.** H1.8 suppresses hyper-individualization through condensins and topo II. (**A**) Schematic of partial TOP2A depletion to test sensitivity of chromosome individualization to TOP2A levels. (**B**) Quantification of chromosome-associated TOP2A upon partial TOP2A depletion. Each dot represents the mean of TOP2A intensity normalized to DNA intensity of a single chromosome cluster (from one nucleus). The data plotted is median ± 95% CI. (**C**) Percent frequency of DNA clusters categorized as unindividualized nuclei (*Figure 6—figure supplement 1C*) upon partial TOP2A depletion. Mean and SEM from three independent experiments. Each dot represents the percentage of unindividualized chromosome clusters in the indicated condition in an independent biological replicate. (**D**) Representative images of nuclei using DNA (Cy5-dUTP) showing the clustering phenotype in the

*Figure 7 continued on next page*

*Figure 7 continued*

indicated conditions. Scale bar, 10 μm. (**E**) Quantification of the three-dimensional surface area of the chromosome clusters in (**D**) normalized to DNA (Cy5-dUTP) signal. Each gray, magenta, green, or orange dot represents the normalized surface area of single nucleus or chromosome cluster, and each black square represents the median surface area of a single experiment. Data plotted is mean and SEM of four independent experiments. (**F**) Quantification of TOP2A immunofluorescence intensity normalized to the DNA signal for the indicated conditions. Each gray or magenta dot represents the average signal intensity of a single chromosome cluster (from one nucleus). Each gray, magenta, orange, or gray dot represents the median signal intensity from a single experiment. Mean and SEM of the median of four independent experiments are also shown. (**G**) Representative images of nuclei using DNA (Cy3-dUTP) showing the clustering phenotype in the indicated conditions. Scale bar, 10 μm. (**H**) Quantification of the three-dimensional surface area of the chromosome clusters in (**G**) normalized to DNA (Cy3-dUTP) signal. Each black open circle represents the normalized surface area of single nucleus or chromosome cluster. Data plotted is median and 95% CI. The p-values in (**C**) and (**F**) were calculated by an unpaired Student's t-test, and the p-values in (**H**) were calculated by a two-tailed Mann–Whitney U-test. The number of nuclei imaged in each condition for each experiment in (**E**), (**F**), and (**H**) is indicated above the figure.

The online version of this article includes the following figure supplement(s) for figure 7:

**Source data 1.** Source data for all the figures in *Figure 7* and its figure supplement.

**Figure supplement 1.** H1.8 suppresses over-individualization through condensins and topo II.

*Goloborodko et al., 2016b*). Similar DNA loop shortening accompanied with increased condensin I loading was also reported on integrated fission yeast genome DNA segments in mouse and human chromosomes (*Fitz-James et al., 2020*). Reduced condensin loading due to mutations in condensin I or expression of phosphomimetic H3 mutants in human cells also led to reduced chromosome length (*Elbatsh et al., 2019*). Condensin binding is similarly suppressed by nucleosomes, but loop extrusion proceeds unhindered through sparsely distributed nucleosomes (*Kong et al., 2020*). Since the length of chromosomes with similar condensin I levels is unaffected by the presence of H1.8 (*Figure 3E,F*), H1.8 may increase the loop size by simply reducing the number of condensin molecules on chromatin but may not necessarily inhibit the loop extrusion rate or processivity.

The regulatory mechanisms of TOP2A recruitment to mitotic chromatin are less clear than those of condensins. Similar to condensin, both TOP2A and TOP2B preferentially localize at active and highly transcribed chromatin, indicating a possible preference for nucleosome-free regions (*Canela et al., 2017*; *Thakurela et al., 2013*; *Yu et al., 2017*). We observe that TOP2A levels on chromatin increase upon linker histone depletion in egg extracts and that linker histone inhibits TOP2A binding

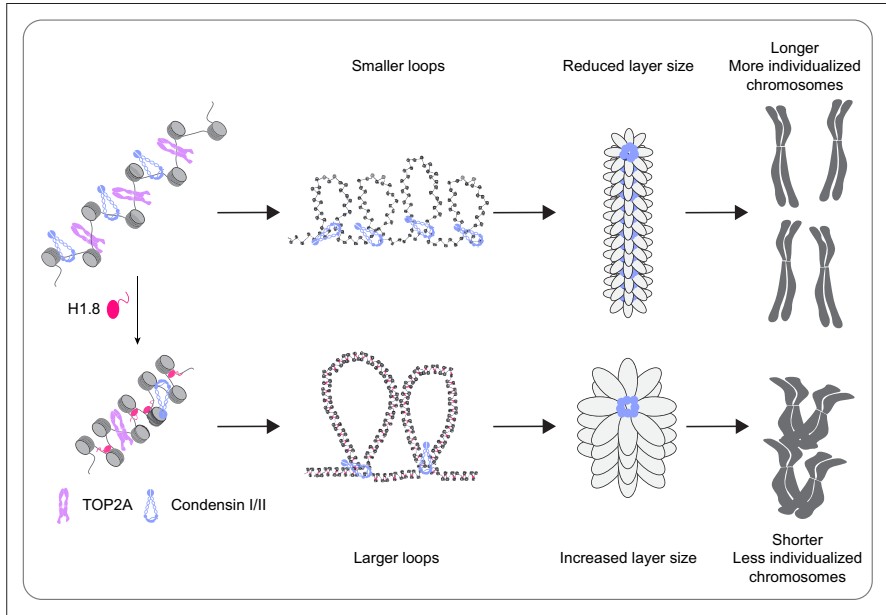

**Figure 8.** A graphical model of how H1.8 controls mitotic chromosome length. In the absence of H1.8, more condensins and topo II bind to more DNA loops of shorter length, resulting in longer and more individualized chromosomes (top). H1.8 limits chromatin levels of condensins and topo II to generate longer and thus fewer DNA loops, resulting in shorter and less individualized chromosomes (bottom).

**Table 1.** Summary table of chromatin levels of condensins, topo II, and chromosome phenotypes. The values of chromatin-bound condensin I (CAP-G), condensin II (CAP-G2), and TOP2A from Figures 6D,E and 7F, Figure 7—figure supplement 1B in the indicated conditions are reported normalized to control (ΔIgG) chromatin. The chromosome lengths from Figure 3 are also normalized to length of control (ΔIgG) chromosomes. Since no individual chromosomes form in ΔCAP-G and ΔH1ΔCAP-GΔCAP-D3 extracts, no length was measurable for those (NA). Chromosome individualization was compared to control (ΔIgG) extracts using a mixture of the surface area (Figure 7) and dilution (Figure 6) assays. + and ++ represent increasing individualization over control extracts, whereas defective extracts fail to individualize chromosomes even by mechanical dispersal.

| | Condensin I | Condensin II | TOP2A | Chromosome length | Chromosome individualization |
|---|---|---|---|---|---|
| ΔH1 | 2.5× | 2× | 3.5× | 1.5× | ++ |
| ΔCAP-G | 0.1× | 1× | 1× | NA | Defective |
| ΔCAP-D3 | 1× | 0.4× | 1× | 1× | Unchanged |
| ΔH1ΔCAP-G | 0.2× | 2× | 3.5× | 0.5× | + |
| ΔH1ΔCAP-D3 | 2.5× | 0.9× | 3.5× | 1.5× | ++ |
| ΔH1ΔCAP-G ΔCAP-D3 | 0.2× | 0.9× | 3.5× | NA | Defective |

to nucleosome arrays in vitro. The preference of topo II for binding linker DNA may also explain the observation of well-spaced TOP2B binding peaks around the well-spaced nucleosomes around the CTCF- binding sites (*Canela et al., 2017*; *Canela et al., 2019*). A recent report has also demonstrated the competition between TOP2A and H1.8 in the absence of nucleosomes (*Shintomi and Hirano, 2021*). Unlike condensin I depletion, TOP2A depletion did not rescue chromosome elongation phenotype of H1.8 depletion. Rather, depletion of topo II leads to elongated chromosomes in vertebrate somatic cell lines and in early embryos of *C. elegans* (*Farr et al., 2014*; *Samejima et al., 2012*; *Ladouceur et al., 2017*; *Nielsen et al., 2020*) and also in *Xenopus* egg extracts (*Figure 3—figure supplement 2*). These data support the proposed structural role for TOP2A in axial compaction of mitotic chromosomes through its C-terminal chromatin-binding domain (*Nielsen et al., 2020*; *Shintomi and Hirano, 2021*; *Lane et al., 2013*).

Our data also suggest that condensins and topo II combine to resolve extensive and persistent interchromosomal topological entanglement during mitotic compaction in *Xenopus* egg extracts (*Figure 5*), but these activities are counterbalanced by H1.8. In HeLa cells, somatic variants of H1 are phosphorylated and evicted along the inter-chromatid axis. This partial H1 eviction in prophase seems to be required for complete decatenation along the chromosome arms (*Krishnan et al., 2017*). This suggests that local enrichment of condensins (particularly condensin II) upon H1 eviction may play a role in sister chromatid decatenation. In contrast, H1.8 binding to nucleosomes is enhanced upon transition from interphase to M phase in *Xenopus* egg extracts (*Arimura et al., 2020*). It might be counterintuitive that condensins and topo II are actively antagonized during mitosis by H1.8 to suppress chromosome individualization, which is a prerequisite for chromosome segregation. Since spindle assembly in oocytes relies on chromatin-induced microtubule nucleation (Heald et al., 1996), keeping chromosomes at metaphase plate might be important for maintaining robust bipolar spindle. As we showed that H1.8 is important for chromosome clustering when spindle assembly is inhibited (*Figure 7*), this microtubule-independent chromosome clustering may be particularly important for the large oocyte and early embryonic cells since it would be difficult to assemble a bipolar spindle onto all the chromosomes once they disperse into the large space of the cytoplasm. Clustering chromosomes through incomplete individualization may also avoid generation of chromosomes that do not associate with the spindle. Such a mechanism would facilitate effective kinetochore attachment during early embryonic cell divisions when the spindle checkpoint cannot be activated by unattached chromosomes (*Mara et al., 2019*; *Gerhart et al., 1984*; *Hara et al., 1980*). Indeed, it was recently shown that paternal and maternal chromosomes cluster at the interface of two pronuclei prior to the first zygotic mitosis after fertilization to facilitate rapid and efficient kinetochore microtubule attachment in human and bovine embryos (*Cavazza et al., 2021*). Another possible reason for the suppressed

individualization is related to the fact that oocyte chromosomes completely lose cohesion from arms at the end of meiosis I, while maintaining sister chromatid cohesion only at the centromeres (*Lister et al., 2010*). Normally, this centromeric cohesion is critical for supporting the kinetochore tension to establish bipolar attachment. During long natural arrest at meiotic metaphase II, these centromeres undergo cohesion fatigue, where centromeres prematurely separate. However, proper segregation may still be accomplished due to apparent inter-chromatids DNA linkages (*Gruhn et al., 2019*). Resolution of these DNA linkages may be prevented by H1.8-mediated suppression of condensin and TOP2A. However, our data do not eliminate the possibility that H1.8 plays a condensin and TOP2A-independent role in regulating chromosome individualization. Regulation of Ki-67, which coats the surface of chromosomes, may be a good candidate of H1-mediated regulation (*White et al., 2016*; *Gibson et al., 2019*; *Cuylen et al., 2016*).

Linker histones are a dynamic component of chromatin (*Misteli et al., 2000*). Linker histone occupancy varies widely (*Woodcock et al., 2006*) and can be controlled by both the linker histone variant and their post-translational modifications (*Th'ng et al., 2005*; *Christophorou et al., 2014*; *Hergeth and Schneider, 2015*). We expect that changing the H1 stoichiometry on chromatin by changing the amount, subtype, and/or affinity (e.g., through post-translational modifications) of H1 affects the length and individualization of chromosomes through regulating the levels of condensins and topo II. In *Xenopus* egg extracts, somatic histone H1 fails to rescue chromosome elongation phenotype of H1.8 depletion since highly abundant importin β sequesters the somatic H1 and suppresses its binding to mitotic chromosomes (*Freedman and Heald, 2010*). It has been shown that the sperm male pronucleus dynamically interacts with microtubules during its dramatic nuclear reorganization upon exposure to the egg cytoplasm (*Xue et al., 2013*), during which H1.8 suppresses chromosome fragmentation mediated by microtubule-dependent force (*Xiao et al., 2012*). Thus, it is possible that H1.8 is tuned to provide chromatin with physical strength of bulk chromatin rather than facilitating chromosome individualization during the unique fertilization process. Linker histones also serve important interphase roles through regulation of transcription (*Izzo et al., 2008*), and the epigenetic landscape of the chromatin determines the linker histone variant on chromatin (*Th'ng et al., 2005*; *Parseghian et al., 2001*; *Izzo et al., 2013*). Since H1.8 competitively inhibits condensin binding in mitosis, it is tempting to speculate that linker histones also inhibit condensin II and cohesin binding in interphase. We suggest that local and global regulation of chromatin structure and function can be regulated by controlling differential expression of linker histone H1 variants and their modifications not just through promoting inter-nucleosomal interactions but also by controlling accessibility of SMC proteins and topo II.

# Materials and methods

**Key resources table**

| Reagent type (species) or resource | Designation | Source or reference | Identifiers | Additional information |
|---|---|---|---|---|
| Cell line (*Spodoptera frugiperda*) | SF9 insect cells | Gibco | 11496015 | |
| Cell line (*Trichoplusia ni*) | High Five insect cells | Gibco | B85502 | |
| Cell line (*Pichia pastoris*) | Yeast cells | Invitrogen/Thermo Fisher Scientific *Ryu et al., 2010* | C18100 | GS115 |
| Biological sample (*Xenopus laevis*) | *Xenopus* | NASCO | LM00531 RRID:XEP_Xla100 | Female, adult frogs |
| Biological sample (*Xenopus laevis*) | *Xenopus* | NASCO | LM00715 RRID:XEP_Xla100 | Male, adult frogs |
| Antibody | Anti-H3 (rabbit polyclonal) | Abcam | RRID:AB_302613 | WB (1 µg/ml) |
| Antibody | Anti-H2B (rabbit polyclonal) | Abcam | RRID:AB_302612 | WB (1 µg/ml) |
| Antibody | Anti α-tubulin (mouse monoclonal) | Sigma-Aldrich | RRID:AB_477593 | WB (1:10000) |
| Antibody | Anti-H1.8 (rabbit polyclonal) | *Jenness et al., 2018* | RU1974 | WB (1 µg/ml) |

*Continued on next page*

*Continued*

| Reagent type (species) or resource | Designation | Source or reference | Identifiers | Additional information |
|---|---|---|---|---|
| Antibody | Anti-TOP2A (rabbit polyclonal) | *Ryu et al., 2010* | NA | WB (2 μg/ml) IF (1 μg/ml) |
| Antibody | Anti-CAP-G (rabbit polyclonal) | *Zierhut et al., 2014* | RU1008 | WB (2 μg/ml) IF (2 μg/ml) |
| Antibody | Alexa 488-anti-CAP-G (rabbit polyclonal) | This study | NA | IF (4 μg/ml), refer to 'Antibodies' section in Methods |
| Antibody | Anti-CAP-D2 (rabbit polyclonal) | *Hirano et al., 1997* | NA | WB (2 μg/ml) |
| Antibody | Anti-CAP-G2 (rabbit polyclonal) | Gift from S. Rankin | OMRF195 | WB (4 μg/ml) IF (4 μg/ml) |
| Antibody | Anti-CAP-D3 (rabbit polyclonal) | This study | RU2042 | WB (2 μg/ml), refer to 'Antibody production' section in Methods |
| Antibody | Anti- CENP-A (rabbit polyclonal) | *Wynne and Funabiki, 2015* | NA | IF (4 μg/ml) |
| Antibody | IRDye 680 LT anti-mouse IgG(H + L) (goat polyclonal) | LI-COR Biosciences | RRID:AB_2687826 | WB (1:10,000) |
| Antibody | IRDye 680 LT anti-rabbit IgG(H + L) (goat polyclonal) | LI-COR Biosciences | RRID:AB_621841 | WB (1:10,000) |
| Antibody | IRDye 800 CW anti-mouse IgG(H + L) (goat polyclonal) | LI-COR Biosciences | RRID:AB_621842 | WB (1:10,000) |
| Antibody | IRDye 800 CW anti-rabbit IgG(H + L) (goat polyclonal) | LI-COR Biosciences | RRID:AB_2687826 | WB (1:10,000) |
| Antibody | IRDye 800 CW anti-mouse IgG(H + L) (goat polyclonal) | LI-COR Biosciences | RRID:AB_621843 | WB (1:10,000) |
| Antibody | Anti-rabbit Alexa 555 (goat polyclonal) | Thermo Fisher Scientific | RRID:AB_141784 | IF (1:1000) |
| Antibody | Anti-rabbit Alexa 555 (goat polyclonal) | Jackson Immunoresearch | RRID:AB_2338079 | IF (1:250) |
| Antibody | Anti-rabbit Alexa 488 F(ab')2 fragment (goat polyclonal) | LifeScience Technologies | RRID:AB_142134 | IF (1:1000) |
| Chemical compound, drug | Nocodazole | Sigma-Aldrich | M1404 | 10 μg/ml |
| Chemical compound, drug | ICRF-193 | Santa Cruz Biotechnology | sc-200889 | 50/500 μM |
| Other | Dynabeads-Protein A | Thermo Fisher Scientific | 100-08D | 250 ng antibody/1 μl beads |
| Other | Dynabeads-M280 Streptavidin | Thermo Fisher Scientific | 11206D | NA |
| Software, algorithm | MATLAB | MathWorks | R2019A | NA |
| Peptide, recombinant protein | DpnII | NEB | R0543 | |
| Peptide, recombinant protein | DNA polymerase I, large (Klenow) fragment | NEB | M0210S | |
| Peptide, recombinant protein | T4 DNA ligase 1 U/μl | Invitrogen | 15224090 | |
| Peptide, recombinant protein | T4 DNA polymerase | NEB | M0203L | |
| Peptide, recombinant protein | T4 polynucleotide kinase | NEB | M0201 | |

*Continued on next page*

*Continued*

| Reagent type (species) or resource | Designation | Source or reference | Identifiers | Additional information |
|---|---|---|---|---|
| Peptide, recombinant protein | Biotin-14-dATP | Invitrogen | 19524016 | |
| Commercial assay or kit | TruSeq Nano DNA Sample Prep Kit | Illumina | 20015964 | |
| Peptide, recombinant protein | Klenow fragment (3′ → 5′ exo-) | NEB | M0212L | |

## Experimental model and subject details

### X. laevis frogs

Animal husbandry and protocol (20031) approved by institutional animal care and use committee (IACUC) of the Rockefeller University were followed. Mature female pigmented *X. laevis* frogs (NASCO-LM00535MX) and male frogs (NASCO-LM00715) were maintained in a temperature-controlled room (16–18°C) using a recirculating water system at The Rockefeller Comparative BioScience Center (CBC). Frogs were temporarily moved to a satellite facility for ovulation.

## Methods

### Antibodies

H3 was detected with ab1791 (Abcam; 1 µg/ml for western blots). H2B was detected with ab1790 (Abcam; 1 µg/ml for western blots). α-tubulin was detected with T9026 (Sigma; 1:10,000 for western blots).

H1.8 was detected using anti-H1M (H1.8) antibody (*Jenness et al., 2018*, 1 µg/ml for western blots). Anti-TOP2A was a gift from Y. Azuma (*Ryu et al., 2010*; 1 µg/ml for western blots and IF). Anti-CAPD2 was a gift from T. Hirano (*Hirano et al., 1997*; 2 µg/ml for western blots). Anti-CAPG2 was a gift from S. Rankin (4 µg/ml for IF, 2 µg/ml for western blots). xCAP-G (*Zierhut et al., 2014*) and xCAP-D3 custom antibodies were used at 2 µg/ml for western blots and 1 µg/ml for IF (for xCAP-G only). CENP-A was detected using an antibody against N-terminal 50 amino acids of CENP-A (*Wynne and Funabiki, 2015*) and used at 4 µg/ml for IF. xCAP-G antibody was also conjugated to Alexa488 using the Alexa488-NHS ester (Thermo Fisher Scientific) using the manufacturer's instructions. The conjugated antibody was purified using the Sephadex G-25 in PD-10 desalting column (Cytiva). The conjugated antibody was dialyzed into PBS + 50% glycerol and stored in aliquots at –80 °C after freezing with liquid nitrogen. The labeled antibody was used at 4 µg/ml for IF.

IRDye 680LT goat anti-mouse IgG (H + L), IRDye 680LT goat anti-rabbit IgG (H + L), IRDye 800CW goat anti-mouse IgG (H + L), and IRDye 800CW goat anti-rabbit IgG (H + L) were used at 1:15,000 (LI-COR Biosciences) dilution for western blots. Alexa 488, Alexa 555, and Alexa647 conjugated secondary antibodies (Jackson Immunoresearch) were used for immunofluorescence.

### Antibody production

xCAP-D3 C-terminal peptide (CRQRISGKAPLKPSN) was synthesized at The Rockefeller University Proteomics Resource Center. The peptide was then coupled to keyhole limpet hemocyanin protein according to the manufacturer's protocol (Thermo Fisher Scientific) and used to immunize rabbits (Cocalico Biologicals). Antibody was purified from the immunized rabbit sera using affinity purification against the same peptide coupled to SulfoLink resin (Thermo Fisher Scientific). The antibody was dialyzed into PBS + 50% glycerol and stored with the addition of 0.05% sodium azide.

### *Xenopus* egg extracts and immunodepletion

Cytostatic factor (CSF)-arrested *X. laevis* egg extracts were generated as previously described (*Murray, 1991*). To generate replicated mitotic chromosomes, 0.3 mM $CaCl_2$ was added to CSF-arrested extracts containing *X. laevis* sperm to cycle the extracts into interphase at 20 °C. 90 min after adding $CaCl_2$, half the volume of fresh CSF extract and 40 nM of the non-degradable cyclin BΔ90 fragment were added to interphase extracts to induce mitotic entry (*Holloway et al., 1993*; *Glotzer et al., 1991*). After 60 min of incubation, extracts were processed for morphological and biochemical

assessments. For all experiments involving immunofluorescence, 10 nM nocodazole was added along with the cyclin BΔ90.

For immunodepletions of 50–100 µl extracts, antibodies were conjugated to Protein-A coupled Dynabeads (Thermo Fisher Scientific) at 250 µg/ml beads, either at room temperature for 60 min or overnight at 4 °C. Mock (IgG) and H1.8 (H1) antibody beads were crosslinked using 4 mM BS₃ (Thermo Fisher Scientific) at room temperature for 45 min and quenched using 10 mM Tris-HCl (Sigma). All antibody beads were washed extensively using Sperm Dilution Buffer (SDB; 10 mM HEPES, 1 mM MgCl₂, 100 mM KCl, 150 mM sucrose) and separated from the buffer using a magnet before addition of extract. H1.8 depletions (ΔH1) were performed with two 45 min rounds of depletion at 4 °C using 2 volumes of antibody-coupled beads for each round. For condensin I and condensin II depletions, 1.5–2 volumes of xCAP-G or xCAP-D3 antibody-coupled beads were used in a single round for depletion for 60 min at 4 °C. For double depletion of condensin I and II, a single round of depletion using 1.5 volume each of xCAP-G and xCAP-D3 antibody-coupled beads was performed. For TopoII depletions (ΔTOP2A), a single round of depletion was performed using 1.2 volume of anti-TopoIIα coupled antibody beads for 60 min at 4 °C. After the incubations, the beads were separated using a magnet.

## Western blots

For total egg extract samples, 1 µl sample was added to 25 µl 1× sample buffer (50 mM Tris-HCl pH 6.8, 2% SDS, 10% glycerol, 2.5% β-mercaptoethanol) and boiled for 10 min. Samples were spun at 8000 rpm for 3 min before gel electrophoresis and overnight transfer at 4 °C. Blotting membranes were blocked with 4% powdered skim-milk (Difco). Primary and secondary antibodies were diluted in LI-COR Odyssey blocking buffer-PBS (LI-COR Biotechnology). Western blots were imaged on a LI-COR Odyssey. Quantifications were done using ImageJ.

## Hi-C

### Standard samples

$10^6$ *X. laevis* sperm nuclei were added to 150 µl interphase extract and allowed to replicate at 21 °C for 90 min. The extracts were cycled back into mitosis by adding 100 µl CSF extract, 40 nM of the non-degradable cyclin BΔ90 and 10 µM nocodazole (Sigma). After 60 min at metaphase, the samples were diluted into 12 ml of fixing solution (80 mM K-PIPES pH 6.8, 1 mM MgCl₂, 1 mM EGTA, 30% glycerol, 0.1% Triton X-100, 1% formaldehyde) and incubated at room temperature with rocking for 10 min. The samples were then quenched with 690 µl 2.5 M glycine for 5 min at room temperature. The samples were then placed on ice for 15 min and then centrifuged at 6000 g at 4 °C for 20 min. The pellet was then resuspended in 1 ml ice-cold DPBS. The tube was then centrifuged again at 13,000 g for 20 min at 4 °C. The buffer was aspirated, and the pellet was frozen in liquid nitrogen and then stored at –80 °C.

### Dispersed chromosome samples

The metaphase chromosome samples were prepared as above, but nocodazole was omitted. The metaphase extracts were diluted by adding 1.2 ml chromosome dilution buffer (10 mM K-HEPES pH 8, 200 mM KCl, 0.5 mM EGTA, 0.5 mM MgCl₂, 250 mM sucrose) and incubated at room temperature for 8 min. 6 ml fixation buffer (5 mM K-HEPES pH 8, 0.1 mM EDTA, 100 mM NaCl, 2 mM KCl, 1 mM MgCl₂, 2 mM CaCl₂, 0.5% Triton X-100, 20% glycerol, 1% formaldehyde) was added to the tube, mixed by rotation 10 min at room temperature. 420 µl 2.5 M glycine was added to quench the formaldehyde, and the mixture was incubated for 5 min at room temperature. The samples were then placed on ice for 15 min and then centrifuged at 6500 g at 4 °C for 20 min. The pellet was then resuspended in 1 ml ice-cold DPBS. The tube was then centrifuged again at 13,000 g for 20 min at 4 °C. The buffer was aspirated, and the pellet was frozen in liquid nitrogen and then stored at –80 °C.

Two biological replicates were performed for each sample, and they confirmed similar behavior among the replicates.

### Library prep and sequencing

Hi-C protocol was performed as previously described (*Belaghzal et al., 2017*), with the exception that cell disruption by douncing was omitted. Briefly, pellets were digested by DpnII overnight at 37 °C prior to biotin fill-in with biotin-14-dATP for 4 hr at 23 °C. After ligation at 16 °C for 4 hr, crosslinking

was reversed by proteinase K at 65 °C overnight. Purified ligation products were sonicated with 200 bp average size, followed by 100–350 bp size selection. End repair was performed on size-selected ligation products, prior to purifying biotin tagged DNA fragments with streptavidin beads. A-tailing was done on the purified DNA fragments followed by Illumina Truseq adapter ligation. Hi-C library was finished by PCR amplification and purification to remove PCR primers. Final library was sequenced on Illumina HiSeq 4000 with PE50.

## Hi-C data processing

Hi-C fastq files were mapped to the *X. laevis* 9.2 genome with the distiller-nf pipeline (https://github.com/open2c/distiller-nf, *Flyamer, 2021*). The reads were aligned with bwa-mem, afterwards duplicate reads were filtered out. These valid pair reads were aggregated in genomic bins of 10, 25, 50, 100, 250, and 500 kb using the cooler format (*Abdennur and Mirny, 2020*). Cooler files were balanced using Iterative balancing correction (*Imakaev et al., 2012*), ignoring first two diagonals to avoid artifacts within the first bin such as re-ligation products. Contact heatmaps from balanced cooler files were viewed and exported with Higlass (*Kerpedjiev et al., 2018*).

## Contact probability (*P(s)*) and derivatives

Contacts probability was calculated by contact frequency (*P*) as function of genomic distance (*s*). Interaction pairs were selected for genomic distance from 1 kb till 100 Mb binned at log-scale. Within each genomic bin, observed numbers of interactions were divided by total possible number of interactions within the bin. Distance decay plots were normalized by total number interactions, and derivative plots were made from corresponding *P(s)*. The derivative plots plotted in *Figure 4C and D*, *Figure 4—figure supplement 1D* were drawn using LOESS smoothing.

## Immunofluorescence

Immunofluorescence was performed according to previously published protocols (*Desai et al., 1998*). 10 µl metaphase extracts containing chromosomes were diluted into 2 ml of fixing solution (80 mM K-PIPES pH 6.8, 1 mM $MgCl_2$, 1 mM EGTA, 30% glycerol, 0.1% Triton X-100, 2% formaldehyde) and incubated at room temperature for 7 min. The fixed chromosomes were then laid onto a cushion (80 mM K-PIPES pH 6.8, 1 mM $MgCl_2$, 1 mM EGTA, 50% glycerol) with a coverslip placed at the bottom of the tube and centrifuged at 5000 g for 15 min at 18 °C in a swinging bucket rotor. The coverslips were recovered and fixed with methanol (–20 °C) for 4 min. The coverslips were then blocked overnight with antibody dilution buffer (50 mM Tris-Cl pH 7.5, 150 mM NaCl, 2% BSA). Primary and secondary antibodies were diluted in antibody dilution buffer and sealed in Prolong Gold AntiFade mounting media (Thermo Fisher Scientific).

For coverslips stained with Alexa488-anti-CAP-G antibody (*Figures 1C, D, 3E and 6D*, *Figure 3—figure supplements 1C and 2D*, *Figure 4—figure supplement 2B* and Figure 6D and E), coverslips stained with primary and secondary antibodies were washed three times with PBS-T (1× PBS + 0.5% Tween-20). Then, they were blocked with 100 µg/ml rabbit IgG or 30 min and were incubated with Alexa488-anti-xCAP-G antibody without any washing steps in between. The coverslips were then washed three times with PBS-T and then sealed in Prolong Gold AntiFade mounting media (Thermo Fisher Scientific).

## Chromosome individualization

Chromosomes from each nucleus often remain clustered in metaphase crude egg extracts. To disperse these clustered chromosomes, extracts containing chromosomes were diluted following a method described before with some modifications (*Funabiki and Murray, 2000*). 40 µl Chromosome Dilution Buffer (10 mM K-HEPES pH 8, 200 mM KCl, 0.5 mM EGTA, 0.5 mM $MgCl_2$, 250 mM sucrose) was added to 10 µl metaphase extract containing chromosomes and incubated at room temperature for 8 min. 200 µl fixation buffer (5 mM K-HEPES pH 8, 0.1 mM EDTA, 100 mM NaCl, 2 mM KCl, 1 mM $MgCl_2$, 2 mM $CaCl_2$, 0.5% Triton X-100, 20% glycerol, 2% formaldehyde) was added to the tube and incubated for 10 min at room temperature. The samples were laid over a cushion (5 mM K-HEPES pH 8, 0.1 mM EDTA, 100 mM NaCl, 2 mM KCl, 1 mM $MgCl_2$, 2 mM $CaCl_2$, 50% glycerol) with a coverslip placed under the cushion and centrifuged at 7000 g for 20 min at 18 °C in a swinging bucket rotor. The coverslips were recovered and fixed with ice-cold methanol for 4 min, washed extensively and blocked

overnight with antibody dilution buffer (50 mM Tris-Cl pH 7.5, 150 mM NaCl, 2% BSA). CENP-A immunofluorescence was performed on these coverslips for *Figures 5D, E and 6B, C*, *Figure 6—figure supplement 1A and B*.

## Chromosome purification

One volume of metaphase extracts with ~3000/µl sperm nuclei was diluted into 3 volumes of DB2 (10 mM K-HEPES, 50 mM β-glycerophosphate, 50 mM NaF, 20 mM EGTA, 2 mM EDTA, 0.5 mM spermine, 1 mM phenylmethylsulfonyl fluoride, 200 mM sucrose) and laid over 1 ml cushion (DB2 with 50% sucrose). The tube was centrifuged in a swinging bucket rotor at 10,000 g for 30 min at 4 °C. Most of the cushion was aspirated and the pellet was resuspended in the remaining solution and transferred to a fresh tube. The sample was centrifuged again at 13,000 g for 15 min at 4 °C. The pellet was then resuspended in 1× sample buffer and boiled for 10 min before being subject to gel electrophoresis.

## Image acquisition and analysis

All the quantitative immunofluorescence imaging and some of the spindle imaging was performed on a DeltaVision Image Restoration microscope (Applied Precision), which is a wide-field inverted microscope equipped with a pco.edge sCMOS camera (pco). The immunofluorescence and surface area measurement samples were imaged with z-sections of 200 nm width with a 100× (1.4 NA) objective and were processed with a iterative processive deconvolution algorithm using the Soft-WoRx (Applied Precision). The dispersed chromosomes imaged for length measurements and chromosome individualization were imaged in five 1 µm z-sections with a 63× (1.33 NA) silicone oil objective.

More than 20 nuclei imaged for immunofluorescence and three-dimensional surface area quantification for most of the experiments for each condition. The differences in each experiment were analyzed using a two-tailed Mann–Whitney $U$-test. The data in *Figures 1D and G, 6D and E and 7E and F*, *Figure 7—figure supplement 1B* were combined data from multiple experiments, where the data were normalized to the control (ΔIgG) levels, whose medians were normalized to 1 for all conditions (*Lord et al., 2020*). The aggregate data were then analyzed using an unpaired Student's $t$-test.

For all the immunofluorescence quantifications, the maximum intensity single slice was selected, background subtraction was performed, and average intensities were calculated on a mask generated using the DNA signal. The analysis was performed using custom MATLAB (MathWorks) code available at https://github.com/pavancss/PC2021_microscopy, copy archived at swh:1:rev:bda651f-09c81eb4233b0bd078e9180166bf6d6ad (*Choppakatla, 2021*).

For surface area measurements, images were interpolated into stacks of 67 nm width. A surface mask was built in three-dimensional space and surface area and DNA signal was calculated using the regionprops3 MATLAB function. Only large objects (>10,000 pixel$^3$) were analyzed to compare significant fractions of each nucleus. The analysis was done automatically using custom MATLAB (MathWorks) code available at https://github.com/pavancss/PC2021_microscopy.

For the CENP-A foci counting in *Figures 5E and 6C*, *Figure 5—figure supplement 1E*, and *Figure 6—figure supplement 1A,B*, DNA and CENP-A were segmented by Otsu's thresholding algorithm. Each independent object in a binarized DNA image was treated as a single chromosomal mass and CENP-A foci were counted in each mass. This was done using custom MATLAB code available at https://github.com/pavancss/PC2021_microscopy.

For the categorization of unindividualized chromosomes in *Figure 6—figure supplement 1D* and *Figure 7C*, a large area of a coverslip was imaged in panels and all the observed DNA masses were counted and categorized as in *Figure 6—figure supplement 1C* in an unblinded fashion. The numbers of DNA masses counted in each condition in *Figure 7C* were: ΔIgG-100% TOP2A (201, 128, 146), ΔIgG-25% TOP2A (155, 177, 144), ΔH1-100% TOP2A (447, 135, 208), and ΔH1-25% TOP2A (232, 187, 171).

For chromosome length measurements, >54 chromosomes were measured in each experiment to ensure that a relatively even sampling of the 18 different chromosomes of each sperm nucleus was possible. Chromosome length measurements were done by manually tracing the chromosomes on a single maximum intensity slice in ImageJ 1.52 p.

## Mononucleosomes and nucleosome arrays

Nucleosome arrays were prepared as previously noted (*Guse et al., 2011*; *Zierhut et al., 2014*). The plasmid pAS696, which contains 19 repeats of the Widom 601 nucleosome position sequence (*Lowary and Widom, 1998*), was digested with EcoRI, XbaI, HaeII, and DraI. The fragment containing the array was isolated using polyethylene glycol-based precipitation. The ends of the DNA fragment were filled in with dATP, dGTP, dCTP, and Bio-16-dUTP (Chemcyte) using Klenow DNA polymerase (NEB) and purified using Sephadex G-50 Nick columns (Cytiva Biosciences).

Mononucleosomal DNA were prepared by digesting pAS696 using AvaI. The 196 bp fragment was isolated using polyethylene glycol-based precipitation. The ends of the fragment were filled in with dATP, dGTP, dTTP, and Alexa647-aha-dCTP (Thermo Fisher Scientific) using Klenow DNA polymerase (NEB) and purified using Sephadex G-50 Nick columns (Cytiva Biosciences).

For nucleosome deposition, 10 μg of DNA arrays or mononucleosomal DNA was mixed with equimolar amount of *X. laevis* H3-H4 tetramer and twice equimolar amount of *X. laevis* H2A-H2B dimers in 1× TE with 2 M NaCl. The mixture was added into in a Slide-A-Lyzer dialysis cassette (Thermo Fisher Scientific) and placed into 500 ml high salt buffer (10 mM Tris-Cl pH 7.5 @ 4 °C, 2 M NaCl, 1 mM EDTA, 5 mM β-mercaptoethanol, 0.01% Triton X-100). Salt was reduced in a gradient by pumping in 2 l of low salt buffer (10 mM Tris-Cl pH 7.5 at 4 °C, 100 mM NaCl, 1 mM EDTA, 5 mM β-mercaptoethanol, 0.01% Triton X-100) at constant volume at 1 ml/min. The quality of the nucleosome arrays was ascertained by digesting the nucleosome arrays with AvaI overnight in low magnesium buffer (5 mM potassium acetate, 2 mM Tris-acetate, 0.5 mM magnesium acetate, 1 mM DTT, pH 7.9) and electrophoresed in a 5% polyacrylamide gel made in 0.5× TBE (45 mM Tris-borate, 1 mM EDTA). The mononucleosomes were assayed by direct electrophoresis.

## Nucleosome-binding assays

Nucleosome arrays were bound to M280 Streptavidin Dynabeads (Thermo Fisher Scientific) in chromatin bead binding buffer (50 mM Tris-Cl pH 8, 150 mM NaCl, 0.25 mM EDTA, 0.05% Triton X-100, 2.5% polyvinylalcohol) by shaking at 1300 rpm for 3.5 hr. To block the Step tagged condensin complexes from binding the unconjugated streptavidin on the beads during the condensin pull downs, the beads were washed once in chromatin binding buffer and then incubated in 1 mM biotin in chromatin-binding buffer by shaking at 1300 rpm for 1 hr. The beads were then washed with chromatin-binding buffer (50 mM Tris-Cl pH 8, 150 mM NaCl, 0.25 mM EDTA, 0.05% Triton X-100) three times, moved to a new tube, washed twice with SDB (10 mM HEPES, 1 mM MgCl₂, 100 mM KCl, 150 mM sucrose), and split into two tubes. SDB with 0.0008% poly-glutamic acid (Sigma; *Stein and Künzler, 1983*) was mixed with 400 nM recombinant xH1.8 (buffer for control) and incubated for 5 min at room temperature. This mixture was incubated with the beads (half with buffer, half with xH1.8) with rotation at 16 °C. The beads were then washed 1× with SDB and 1× with binding buffer (10 mM HEPES pH 8, 40 mM NaCl, 2.5 mM MgCl₂, 0.5 mM DTT, 0.05% Triton X-100). Beads were washed 2× with binding buffer with indicated assay salt concentration and resuspended in binding buffer with 100 nM recombinant TOP2A, 380 nM human condensin I, condensin I Q loop mutant, or 320 nM condensin II or condensin II Q loop mutant. The beads were rotated at room temperature for 30 min. Total reaction samples were taken, and the beads were washed three times on a magnet in binding buffer and moved to a new tube. The beads were collected on a magnet and resuspended in 1× sample buffer (50 mM Tris-HCl pH 6.8, 2% SDS, 10% glycerol, 2.5% β-mercaptoethanol) and boiled for 5 min. Gel electrophoresis was performed, and the gels were stained with GelCode Blue Stain reagent (Thermo Fisher Scientific).

## Condensin gel shift assays

200 nM Alexa647 labeled 196 bp mononucleosomes were mixed with 0.0008% poly-glutamic acid (Sigma; *Stein and Künzler, 1983*) and half was mixed with 400 nM recombinant xH1.8 in 1× binding buffer (10 mM HEPES pH 8, 50 mM NaCl, 2.5 mM MgCl₂, 5 mM ATP, 0.5 mM DTT, 0.05% Triton X-100) and incubated for 30 min at room temperature. 100 nM of the mononucleosomes with or H1.8 were mixed with the indicated concentration of condensin I in 1× binding buffer at 4 °C for 30 min and subject to electrophoresis onto a 5% polyacrylamide gel in 0.5× TBE at room temperature. The gels were imaged on a LI-COR Odyssey (LI-COR Biotechnology). The binding curves were fitted using GraphPad Prism 8.4.3 using the sigmoidal binding curve option of the nonlinear curve fitting.

## Protein purification

### H1.8

A pET51b vector expressing *X. laevis* H1.8 with an N-terminal Strep-Tag II and C-terminal 6× Histidine-tag was a gift from Rebecca Heald (UC Berkeley). *E. coli* Rosetta2 (DE3 pLysS) cells containing expression plasmids were grown in TBG-M9 media (15 g/l tryptone, 7.5 g/l yeast extract, 5 g/l NaCl, 0.15 g/l MgSO$_4$, 1.5 g/l NH$_4$Cl, 3 g/l KH$_2$PO$_4$, 6 g/l Na$_2$HPO$_4$; 0.4% glucose) at 37 °C until they reach OD ~0.6 and were supplemented with 1 mM isopropylthio-β-galactoside (IPTG) and grown at 18 °C for 14 hr. Cells were collected and resuspend in lysis buffer (1× PBS, 500 mM NaCl, 10 % glycerol, 20 mM imidazole, 0.1% Triton X-100, 10 mM β-mercaptoethanol, 1 mM phenylmethylsulfonyl fluoride, 10 µg/ml leupeptin, 10 µg/ml pepstatin, 10 µg/ml chymostatin). All subsequent steps were carried out at 4 °C. After 30 min incubation, the cell suspension was sonicated and centrifuged at 45,000 g for 45 min at 4 °C. The supernatant was added to Ni-NTA beads (Bio-Rad) and rotated for 60 min. The beads were then washed with Wash Buffer 1 (1× PBS, 20 mM imidazole, 500 mM NaCl, 4 mM β-mercaptoethanol, 10 mM ATP, 2.5 mM MgCl$_2$, cOmplete EDTA-free protease inhibitor cocktail; Roche). The beads were eluted with NTA elution buffer (1× PBS, 400 mM imidazole, 500 mM NaCl). The correct fractions were collected and dialyzed into PBS supplement with 500 mM NaCl, concentrated using Amicon Ultra centrifugal filters (10 k cutoff), flash frozen, aliquoted, and stored at –80 °C.

### TopoIIα

*X. laevis* TOP2A tagged with calmodulin-binding protein (CBP) was purified from *Pichia pastoris* yeast as reported (**Ryu et al., 2010**) with some modifications. *P. pastoris* integrated with a CBP tagged TOP2A cassette under the influence of an alcohol oxidase (AOX) promoter (a gift from Yoshiaki Azuma) were grown in BMGY media (1% yeast extract, 2% peptone, 100 mM potassium phosphate pH 6, 1.34 % yeast nitrogen base, 4 × 10$^{-5}$ % biotin, 1 % glycerol) containing 50 µg/ml G418 (Thermo Fisher Scientific) at 30 °C until OD ~ 4.0. The cells were collected by centrifugation and split into BMMY media (1% yeast extract, 2% peptone, 100 mM potassium phosphate pH 6, 1.34% yeast nitrogen base, 4 × 10$^{-5}$ % biotin, 0.5% methanol) and grown at 22 °C for 14 hr. The cells were collected, packed into a syringe, and extruded into liquid nitrogen in the form of noodles. These frozen noodles were lysed using a Retsch PM100 cryomill (Retsch) with continuous liquid nitrogen cooling. The cyromilled cells were then resuspended in Lysis Buffer (150 mM NaCl, 18 mM β-glycerophosphate, 1 mM MgCl$_2$, 40 mM HEPES [pH 7.8], 5% glycerol, 0.1% Triton X-100, 1 mM DTT, cOmplete EDTA-free protease inhibitor tablet) and sonicated on ice. The cells were centrifuged at 35,000 g for 45 min at 4 °C. 2 mM CaCl$_2$ was added to the supernatant along with calmodulin-sepharose beads (Strategene) and the mixture was incubated at 4 °C for 120 min. The beads were then washed with ATP-Wash Buffer (Lysis Buffer + 5 mM MgCl$_2$, 2 mM CaCl$_2$, 1 mM ATP), Wash Buffer 1 (Lysis Buffer + 2 mM CaCl$_2$), Wash Buffer 2 (300 mM NaCl, 1 mM MgCl$_2$, 2 mM CaCl$_2$, 20 mM HEPES [pH 7.8], 5% glycerol, 1 mM DTT) and then eluted into elution buffer (300 mM NaCl, 1 mM MgCl$_2$, 5 mM EGTA, 20 mM HEPES [pH 7.8], 5% glycerol, 1 mM DTT).

The eluted protein was then passed through a MonoQ anion exchange column (Cytiva) on an AKTA-FPLC (Cytiva) to separate co-purified DNA. The flowthrough was then digested with TEV protease to cleave the CBP tag and then loaded on a HiTrap Heparin HP column (Cytiva) on an AKTA-FPLC and eluted using a salt gradient of 150 mM NaCl to 1 M NaCl. The selected fractions were then loaded on a Superose 6 gel filtration column (Cytiva) and eluted in freezing buffer (250 mM NaCl, 1 mM MgCl$_2$, 20 mM HEPES pH 7.8, 5% glycerol, 1 mM DTT). The protein was then concentrated and frozen in aliquots at –80 °C.

### Condensins

Human condensin complexes were purified as described previously (**Kong et al., 2020**). Briefly, the five subunits of human condensin I and II, sub-complexes, and Q-loop mutations and were assembled into biGBac vectors (**Weissmann et al., 2016**) to create baculovirus for protein expression in HighFive insect cells. Cell were lysed in condensin purification buffer (20 mM HEPES [pH 8], 300 mM KCl, 5 mM MgCl$_2$, 1 mM DTT, 10% glycerol) supplemented with Pierce protease inhibitor EDTA-free tablet (Thermo Scientific) and Benzonase (Sigma). Cleared lysate was loaded on to a StrepTrap HP (GE), washed with condensin purification buffer, and eluted with condensin purification buffer supplemented with 5 mM Desthiobiotin (Sigma). Protein-containing fractions were pooled, diluted twofold

with Buffer A (20 mM HEPES [pH 8], 5 mM MgCl$_2$, 5% glycerol, 1 mM DTT), loaded on to HiTrap Heparin HP column (GE), washed with Buffer A with 250 mM NaCl, then eluted with buffer A with 500 mM NaCl. Finally, size-exclusion chromatography was performed using condensin purification buffer and a Superose 6 16/70 or increase 10/300 column (GE).

### Mass photometry

All mass photometry data were taken using a Refeyn OneMP mass photometer (Refeyn Ltd). Movies were acquired for 10,000 frames (100 s) using AcquireMP software (version 2.4.0) and analyzed using DiscoverMP software (version 2.4.0, Refeyn Ltd), all with default settings. Proteins were measured by adding 1 µl of stock solution (50 nM) to a 10 µl droplet of filtered buffer (10 mM HEPES pH 8, 2.5 mM MgCl$_2$, 1 mM DTT, 50–300 mM NaCl, 5 mM ATP). Contrast measurements were converted to molecule weights using a standard curve generated with bovine serine albumin (Thermo 23210) and urease (Sigma U7752).

### Mass spectrometry

Sperm chromosomes were purified as previously described (*Funabiki and Murray, 2000*). Briefly, extracts containing 8000/µl sperm nuclei were replicated along with 5 mM biotin-dUTP for 90 min and cycled back into metaphase with the addition of 1 volume of fresh CSF depleted (correspondingly ΔIgG or ΔH1). After 60 min in metaphase, these chromosomes were diluted in 3 volumes of DB (10 mM K-HEPES [pH 7.6], 100 mM KCl, 2 mM EDTA, 0.5 mM EGTA, 0.5 mM spermine, 250 mM sucrose, 1 mM PMSF, and 10 µg/ml each of leupeptin, pepstatin, and chymostatin) and centrifuged through a 60 DB cushion (DB with 60% [w/v] sucrose). The collected chromosome-enriched pellet was then incubated with 15 µl streptavidin-coupled Dynabeads (M280) and rotated at 4 °C for 2 hr. The beads were then collected, washed, and boiled in sample buffer before running on a 6% polyacrylamide gel for 10 min. The gel was stained with Commassie blue, and the protein-containing gel fragments were cut out and processed for mass spectrometry. The mass spectrometry was performed at the Rockefeller University Proteomics Resource Center as previously described (*Zierhut et al., 2014*), but the peptides were queried against the *X. laevis* database (*Wühr et al., 2014*) using the MaxQuant software (Max-Planck Institute).

## Acknowledgements

We thank C Zierhut for providing ΔH3-H4-depleted extracts for Hi-C analysis; S Rankin, T Hirano, Y Azuma for sharing reagents; C Jenness for anti-H1.8 antibodies; JF Martinez and MP Rout for their help with purifying TOP2A; A North, C Rico, and K Cialowicz at Bioimaging Resource Center (BIRC) for help with imaging; C Steckler and H Molina at The Rockefeller Proteomics Core Facility for help with mass spectrometry and analyzing the data; K Mickolajczyk and T Kapoor for help with mass photometry, Y Arimura, R Heald, L Mirny, A Vannini, and C Zierhut and members of the Funabiki lab for helpful discussions. This work was supported by the National Institutes of Health (NIH) grant R35 GM132111 to HF and NIH grant R01 HG003143 to JD. JD is an investigator of the Howard Hughes Medical Institute. AV is supported by a Cancer Research UK Programme Foundation (CR-UK C47547/A21536) and a Wellcome Trust Investigator Award (200818/Z/16/Z). HF is affiliated with Graduate School of Medical Sciences, Weill Cornell Medicine, and Cell Biology Program, the Sloan Kettering Institute. The authors declare that no competing financial interests exist.

## Additional information

### Competing interests

Job Dekker: Reviewing editor, *eLife*. The other authors declare that no competing interests exist.

## Funding

| Funder | Grant reference number | Author |
|---|---|---|
| National Institutes of Health | R35 GM132111 | Hironori Funabiki |
| National Institutes of Health | R01 HG003143 | Job Dekker |
| Cancer Research UK | CR-UK C47547/A21536 | Alessandro Vannini |
| Wellcome Trust | 200818/Z/16/Z | Alessandro Vannini |
| Howard Hughes Medical Institute | Investigator Program | Job Dekker |

The funders had no role in study design, data collection and interpretation, or the decision to submit the work for publication.

## Author contributions

Pavan Choppakatla, Conceptualization, Data curation, Formal analysis, Investigation, Methodology, Resources, Software, Supervision, Validation, Visualization, Writing - original draft, Writing - review and editing; Bastiaan Dekker, Data curation, Formal analysis, Investigation, Methodology, Resources, Software, Visualization, Writing - review and editing; Erin E Cutts, Resources, Writing - review and editing; Alessandro Vannini, Resources, Supervision, Writing - review and editing; Job Dekker, Data curation, Funding acquisition, Methodology, Project administration, Resources, Supervision, Writing - review and editing; Hironori Funabiki, Conceptualization, Funding acquisition, Project administration, Resources, Supervision, Validation, Visualization, Writing - original draft, Writing - review and editing

## Author ORCIDs

Pavan Choppakatla (iD) http://orcid.org/0000-0003-0387-913X
Erin E Cutts (iD) http://orcid.org/0000-0003-3290-4293
Job Dekker (iD) http://orcid.org/0000-0001-5631-0698
Hironori Funabiki (iD) http://orcid.org/0000-0003-4831-4087

## Ethics

This study was performed in strict accordance with the care standards provided by the 8th edition of the Guide for the Care and Use of Laboratory Animals. African clawed frogs, Xenopus laevis, which were maintained and handled according to approved institutional animal care and use committee (IACUC) protocol (20031) of the Rockefeller University, which is an Association for Assessment and Accreditation of Laboratory Animal Care International (AAALAC) accredited research facility.

## Decision letter and Author response

Decision letter https://doi.org/10.7554/eLife.68918.sa1
Author response https://doi.org/10.7554/eLife.68918.sa2

# Additional files

## Supplementary files

• Transparent reporting form

## Data availability

Hi-C sequencing data have been deposited in GEO under an accession code GSE164434. All other data generated or analyzed during this study are included in the manuscript and supporting source data files.

The following dataset was generated:

| Author(s) | Year | Dataset title | Dataset URL | Database and Identifier |
|---|---|---|---|---|
| Choppakatla P, Dekker B, Cutts EE, Vannini A, Dekker J, Funabiki H | 2021 | Linker histone H1.8 inhibits chromatin-binding of condensins and DNA topoisomerase II to tune chromosome compaction and individualization | https://www.ncbi.nlm.nih.gov/geo/query/acc.cgi?acc=GSE164434 | NCBI Gene Expression Omnibus, GSE164434 |

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
