## [Decision Letter]

**Acceptance summary:**

This paper takes advantage of the *Xenopus* egg cell free system, combining classical immunofluorescence assays with chromosome conformation (Hi-C) analyses to elucidate the contribution of linker histone H1 to mitotic chromosome organization. The authors find that linker histone H1.8 limits the association of condensin and topoisomerase II to control both chromosome length and individualization. These effects of histone HI on larger scale chromosome organisation may be important for clustering chromosomes for segregation in cells with a large cytoplasm, such as oocytes.

**Decision letter after peer review:**

Thank you for submitting your article "Linker histone H1.8 inhibits chromatin-binding of condensins and DNA topoisomerase II to tune chromosome length and individualization" for consideration by *eLife*. Your article has been reviewed by 3 peer reviewers, including Adèle L Marston as the Reviewing Editor and Reviewer #1, and the evaluation has been overseen by Jessica Tyler as the Senior Editor.

Essential revisions:

1) The authors conclude from their study that Condensins and topo II "functionalities are antagonized by H1.8 to tune chromosome length and prevent hyper-individualization". This interpretation comes from the removal of histone H1.8 from extracts/chromatin and the authors suggest that this is due to a direct competition of H1.8 with Condensins and topo II. However, they show in Figure 1—figure supplement 2 that the abundance of several chromatin proteins is also increased when depleting histone H1.8. Since one of those proteins is histone H1.3, it is unclear how the author can directly ascribe the increased binding of topo II and Condensins on chromosome to the removal of H1.8. Could it not be that removal of H1.8 leads to enrichment of H1.3 and that the latter histone has topo II/Condensin-loading properties on chromosomes? In fact, any of the proteins whose abundance increases due to the loss of H1.8 could act as a topo II/Condensin-loader on chromosomes, which would also explain the authors observations.

2) The purified Condensin complexes used in the DNA/nucleosome binding experiments have major issues with its subunit stoichiometry. Inspection of band patterns in the Condensin input gels reveals that the purified preparations used in this study fluctuate between nearly stoichiometric (Figure 2—figure supplement 2B), to partly stoichiometric (Figure 2—figure supplement 2A) to severely unbalanced (Figure 2A,B; Figure 2—figure supplement 2C). In all cases, the 8S (CAP-E/C) subcomplex of Condensin is overrepresented relative to the 11S subcomplex (CAP-G/H/D2). This indicates enzyme subunit dissociation during purification. This subunit imbalance is even more dramatic when comparing the DNA-bound (bead) fraction to the inputs in these experiments. The bead fraction is typically pulling down ~10-fold more CAP-E/C than other subunits. This indicate that the DNA/nucleosome binding behavior attributed to the Condensin "complex" is in fact reflective of only the SMC subcomplex (CAP-E/C without CAP-G/H/D2). As a consequence, most or all the DNA/nucleosome array binding experiments need to be repeated with genuine/ stoichiometric Condensin complexes to provide valid conclusions on the DNA binding behavior of this enzyme.

3) Regarding chromosome individualization, the authors show that the percentage of "individualized chromosomes" is the same in mock and H1 depleted extracts (after dilution of the chromosome preparation before fixation). However, the dispersion of the chromosomes (processed for immunofluorescence without dilution) is clearly higher in the latter. What is driving the clustering? Interchromosomal entanglements? If so, shouldn´t the entanglements prevent chromosome individualization upon dilution?

4) Related to point (3), the link between H1, condensin, Top2A and chromosome individualization is not clear. In Figure 6, chromosome individualization is measured by the separation of CENP-A foci, while in Figure 7, the authors examine surface area. These measurements may underlie quite distinct phenomena and so it is unclear what the different measurements in each of the conditions refer to. The authors need to clarify what these differences mean.

5) On the same line, the authors write at the beginning of Discussion: "H1 (…) can also suppress chromosome dispersion that is driven by condensins and topo II that organize the long-range DNA folding". Two issues here:

a) It is not clear which results in the paper actually connect long range DNA folding with the dispersion phenotype.

b) The role of topoII in long range DNA folding is not tested in the paper.

Maybe the contribution of topoII to chromosome length could be tested using the same scheme as in Figure 6G-H and the protocol for observing individualized chromosomes (dilution before fixation)

6) A previous study showed that depletion of condensin I results in complete loss of individualization of replicated chromosomes in *Xenopus* egg extracts (Shintomi and Hirano, Genes Dev 2011), whereas in the current study the authors see only a partial effect (Figure 5C). Could this mean that depletion is not complete (western bloting in Figure 5A suggests this might be the case)? Incomplete depletion would complicate the interpretation of the rescue experiment. Does the amount of protein in the extract and relate this to the extent of chromosomal condensation observed? The authors need to explain this discrepancy with previously published work.

7) The authors propose that one of histone H1.8 role is to reduce chromosome length (a function achieved by antagonizing topo II/Condensin loading on chromosome). However, histone H1.8 is an embryonic linker histone and cells in early embryos are typically large and less dependent on chromosome size-reduction mechanisms. One would assume that chromosome size should be maximally reduced in small somatic cells and, consequently, shortening of chromosomes should be promoted by somatic linker histone H1 subtypes. In *Xenopus laevis*, histone H1.8 accumulates during oogenesis, persists until the mid-blastula transition, and is then replaced by somatic H1 subtypes. It seems like H1.8 disappears when its putative chromosome shortening function is most needed. The authors did not explain how their results are consistent with this biological context.

*Reviewer #1 (Recommendations for the authors):*

1. Is there a threshold amount of condensin I that needs to be associated with chromosomes for "normal" length. Can the authors partially deplete histone H1 and see a correlation between the amount of condensin and chromosome length?

2. Related to 2. If chromosome individualization is affected in H1 depletion, the authors should be able to observe inter-chromosomal interactions in their Hi-C dataset. Can this be quantified for the different conditions?

*Reviewer #2 (Recommendations for the authors):*

This is a very nice study from a laboratory that has contributed several important papers to the field of mitosis over the years. Experiments were well designed and carefully performed for the most part (with the exception of DNA/nucleosome binding experiments with purified Condensins). The flow of experiments in this manuscript is logical. Overall, I would say this manuscript addresses an important biological question, and this topic has obvious implications for development and genome stability. The manuscript would however benefit from toning down several overinterpretations and would also be improved by providing a more compelling rationale to justify a role for an early embryonic histone in the reduction of chromosome length.

1 – All experiments presented in this study should be performed a minimum of 3 times as this represents a basic requirement for reproducibility in biomedical research. This includes all Hi-C experiments (two replicates are reported) and some nucleosome-binding experiments (Figure 2—figure supplement 3).

2 – There is a missing reference in the following sentence on page 4: Since Condensins prefer to bind nucleosome-free DNA (Kong et al. 2020; Zierhut et al., 2014; Shintomi et al., 2017; Toselli‐Mollereau et al., 2016). This observation was originally made by Piazza et al., 2014, Nat Struct Mol Biol.

*Reviewer #3 (Recommendations for the authors):*

It could be very useful to present a table in the Discussion with a summary of:

1 – the relative amounts of condensins I and II and topo II on DNA , taken from figures 5D, 5E, and Figure 6_supplement1 C.

2 –the phenotype observed in terms of chromosome length, chromosome individualization, and clustering, under the different conditions tested (control, δ H1, deltaH1deltaG, deltaH1deltaD3 and deltaH1deltaGdeltaD3).

This would help understand the co-dependencies in recruitment of the factors under study and the correlation between the different phenotypes.

[Editors' note: further revisions were suggested prior to acceptance, as described below.]

Thank you for resubmitting your work entitled "Linker histone H1.8 inhibits chromatin-binding of condensins and DNA topoisomerase II to tune chromosome length and individualization" for further consideration by *eLife*. Your revised article has been reviewed by 3 peer reviewers, including Adèle L Marston as the Reviewing Editor and Reviewer #1, and the evaluation has been overseen by Jessica Tyler as the Senior Editor.

The manuscript has been improved but there are some remaining issues that need to be addressed, as outlined below:

You will see below that Reviewer #2 has two outstanding issues. Please address these as follows:

1. Please add a sentence to the Results section for the condensin binding assays to qualify that SMC subunits may be retained on the chromatin beads more efficiently than the other condensin subunits. You may also want to speculate why this is the case, as in your rebuttal letter.

2. While three repetitions of every experiment are not insisted on, it is important that the statistical analysis used is appropriate and clearly reported throughout the manuscript. In a number of places in the manuscript, t tests appear to have been carried out where n=2. If/where this is the case, please reanalyse the data using the appropriate statistical test and state this clearly in the figure legend. Where key experiments fail to show significance using these criteria, an additional repeat may be appropriate. For example, the legend of Figure 2 states "The statistical significance of the changes in each immunofluorescence experiment was assessed by a Mann Whitney U-test and the p-values shown in D, E and G are calculated by an unpaired Student's t-test of the aggregate medians (immunofluorescence) or blot intensities of each single experiment." A t-test would not be appropriate for Figure 2D where it appears that two immunofluorescence experiments are presented, if this is indeed what is shown. Please also refer to the comment by reviewer #2.

In addition, please note suggestions for textual changes from Reviewer #1.

*Reviewer #1 (Recommendations for the authors):*

The authors have addressed all of the essential revisions. In addition they have provided convincing responses to all reviewers comments. There are a few places where the text could be clarified:

Line 185. Please explain the Q loop mutant

Line 195 "Although SMC2 and SMC4 appeared to be better retained on nucleosome beads than the non-SMC subunits"

Line 318 "This is consistent with the shorter chromosomes in DH1DCAP-G1 extracts as compared to control…"

Line 362: Please explain ICRF-193.

Line 368: Suggest: To quantify the clustering, we stained the coverslips for CENP-A and used the CENP-A foci (CENP-A doublet is counted as one focus) to measure the number of chromosomes in each chromosome mass. The fraction of chromosomes was measured in small clusters (<4 CENP-A foci, since a few chromosomes can colocalizae stochastically).

Line 383 "better" can be deleted here since the assay has already been introduced.

Line 421 "showed reduced clustering compared to.."

Line 427 Suggest replacing "background" with "extracts"

Line 428 "TOP2A depletion in H1.8 depleted extracts also did not reduce the chromosome spreading even though TOP2A levels on these chromosomes became comparable to that of the control".

*Reviewer #2 (Recommendations for the authors):*

The revised manuscript has been improved by the authors but two issues raised in previous reviews have not been addressed in the latest version of the manuscript.

1 – Condensin stoichiometry: Experiments shown in Figure 2 and associated supplemental figures use Condensin complexes with non-stoichiometric subunit composition. The authors recognize this fact in their rebuttal letter, stating that the "stoichiometry of condensin subunits seems to be changed in DNA-bound fraction in DNA-beads pull-down experiments…" The issue is not convincingly resolved by the mass photometry experiment shown in Figure 2—figure supplement 1 because this experiment does not fully recapitulate conditions of the DNA binding reactions.

The authors suggest problems with stoichiometry are not relevant because Condensin complexes used in their experiments were "purified as reported previously (Kong et al., 2020)" and "are similar to those seen in previous reports in both recombinant yeast condensin and immunoprecipitated *Xenopus* and human condensins (Kong et al., 2020; Ganji et al., 2018; Kimura, Cuvier, and Hirano 2001; Kimura and Hirano 1997)." Examination of Condensin purification gels from Kong et al., shows that the complexes purified by these authors are actually stoichiometric (see Figure 1D in PMID 32445620). The same conclusion can be reached for the yeast complexes purified by the Häring group (see Figure 1C in PMID 28882993). The other two studies cited by the authors (Kimura, Cuvier, and Hirano 2001; Kimura and Hirano 1997) use silver staining to label Condensin subunits, a staining method that most biochemists recognize as non-linear and inappropriate to compare the abundance of different proteins. The contention that subunit imbalances seen in the Condensin complexes used by Choppakatla and colleagues are normal and/or reflect imbalances seen in other purified Condensin complexes cannot be verified in the published literature.

The DNA/nucleosome binding behavior attributed to Condensin is this study is likely reflective of the SMC subcomplex (CAP-E/C without CAP-G/H/D2), not the genuine Condensin 1 enzyme. As a consequence, most or all the DNA/nucleosome array binding experiments need to be repeated with stoichiometric Condensin complexes to provide valid conclusions on the DNA binding behavior of the entire/native enzyme.

2 – Several experiments presented in the revised manuscript have been performed as duplicates, and on one occasion (Figure 3E-F) the results presented come from a single experiment. This reviewer argued previously that all experiments in this study should be performed a minimum of 3 independent times. The authors stated in their rebuttal letter that they "do not see that the level of confidence significantly increases by enforcing every experiment to be done 3 times, statistically speaking." I disagree with this statement. There is ample evidence to show that statistical analyses are less reliable with sample sizes of 2. Many textbooks /reference papers in statistics support this view and, as an example, I would refer the authors to "A biologist's guide to statistical thinking and analysis." The authors are correct, though, in their statement that *eLife* has no explicit publication policy on the minimal number of independent replicates required for publication.

The insistence of the authors to use two independent datasets/experiments for most figures has statistical implications. For example, on some occasions, the authors use a t-test to compare two groups, which is a parametric test. On other occasions, the authors use a Mann Whitney U test, which is a non-parametric test. Parametric testing assumes that the data are normally distributed. With n=2, it is expected that the data would not pass a normality test, hence a Mann-Whitney U test is likely the only appropriate option for most of the figures presented in the manuscript. Separate from these points, when comparing multiple groups on a graph, a One-Way ANOVA is more appropriate than performing multiple independent t-tests. Overall, the authors need to justify their rationale for the choice of statistical analyses performed throughout the manuscript, especially when using only n=2. In many cases, performing an extra experiment to bring datasets to n=3 would likely alleviate the concerns expressed above.

*Reviewer #3 (Recommendations for the authors):*

The authors have clarified most of my doubts and answered carefully to the criticisms raised during revision. The new figures and changes in the text have improved the manuscript and have made it more clear and accessible to readers.

---

## [Author Response]

Essential revisions:1) The authors conclude from their study that Condensins and topo II "functionalities are antagonized by H1.8 to tune chromosome length and prevent hyper-individualization". This interpretation comes from the removal of histone H1.8 from extracts/chromatin and the authors suggest that this is due to a direct competition of H1.8 with Condensins and topo II. However, they show in Figure 1—figure supplement 2 that the abundance of several chromatin proteins is also increased when depleting histone H1.8. Since one of those proteins is histone H1.3, it is unclear how the author can directly ascribe the increased binding of topo II and Condensins on chromosome to the removal of H1.8. Could it not be that removal of H1.8 leads to enrichment of H1.3 and that the latter histone has topo II/Condensin-loading properties on chromosomes? In fact, any of the proteins whose abundance increases due to the loss of H1.8 could act as a topo II/Condensin-loader on chromosomes, which would also explain the authors observations.

Our data show that H1.8 depletion results in increased accumulation of condensins and topo II on mitotic chromatin in *Xenopus* egg extracts (Figure 1) and that H1.8 can directly inhibit binding of condensins and topo II to the linker DNA between nucleosomes in vitro with purified components (Figure 2). Thus, we believe that a direct competition between H1.8 and condensins and topo II for the linker DNA represents a very likely model to explain the chromatin enrichment of condensins and topo II in egg extracts. Other DNA binding proteins also showed some changes in chromatin binding in H1.8 depleted extracts (Figure 1-figure supplement 1E, figure supplement 2, 3), but condensins and topo II are the most abundant chromatin proteins whose chromatin binding is significantly affected by H1.8 depletion. The amount of H1.3 bound to chromatin upon H1.8 depletion was around two orders of magnitude lower than that of condensins and topo II on chromatin (Figure 1-figure supplement 1E) and was 6-fold lower than the residual undepleted H1.8. Therefore, it would be highly unlikely that H1.3 directly stimulates condensin and topo II recruitment unless it can exhibit an unknown catalytic action. If that is the case, it is clearly out of the scope of this manuscript. Interestingly, we now showed that partial depletion of TOP2A did not rescue chromosome elongation phenotype of H1.8 depletion. We believe that this new result would further highlight the unique role of condensin in chromosome elongation.

2) The purified Condensin complexes used in the DNA/nucleosome binding experiments have major issues with its subunit stoichiometry. Inspection of band patterns in the Condensin input gels reveals that the purified preparations used in this study fluctuate between nearly stoichiometric (Figure 2—figure supplement 2B), to partly stoichiometric (Figure 2—figure supplement 2A) to severely unbalanced (Figure 2A,B; Figure 2—figure supplement 2C). In all cases, the 8S (CAP-E/C) subcomplex of Condensin is overrepresented relative to the 11S subcomplex (CAP-G/H/D2). This indicates enzyme subunit dissociation during purification. This subunit imbalance is even more dramatic when comparing the DNA-bound (bead) fraction to the inputs in these experiments. The bead fraction is typically pulling down ~10-fold more CAP-E/C than other subunits. This indicate that the DNA/nucleosome binding behavior attributed to the Condensin "complex" is in fact reflective of only the SMC subcomplex (CAP-E/C without CAP-G/H/D2). As a consequence, most or all the DNA/nucleosome array binding experiments need to be repeated with genuine/ stoichiometric Condensin complexes to provide valid conclusions on the DNA binding behavior of this enzyme.

The human condensin complexes used in the binding assays in Figure 2 were purified as reported previously (Kong et al., 2020). These complexes eluted as a single peak in a gel filtration column, and the varied intensities of the subunits in the complex are similar to those seen in previous reports in both recombinant yeast condensin and immunoprecipitated *Xenopus* and human condensins (Kong et al., 2020; Ganji et al., 2018; Kimura, Cuvier, and Hirano 2001; Kimura and Hirano 1997). Apparent difference in band intensities in the input samples is most likely due to the different numbers of Coomassie dye binding sites among different condensin subunits. In order to confirm that the condensin complexes used in the binding assays were intact full-length complexes, we also performed mass photometry in the binding assay conditions and added these as Figure 2—figure supplement 1. We had also shown that the wildtype human condensin I was functional as it was able to rescue chromosome morphology in condensin I depleted egg extracts (Figure 2-figure supplement 2A). Although stoichiometry of condensin subunits seems to be changed in DNA-bound fraction in DNA-beads pull-down experiments, most likely due to selective loss of non-SMC subunits during bead-washing steps, H1.8 clearly reduced DNA-binding of all condensin subunits. Since we observed H1.8 mediated inhibition of condensin binding by both chromatin pulldowns (Figure 2) and using electrophoresis mobility shift assays (EMSA Figure2-figure supplement 4), we believe that this represents the behavior of full length condensin complexes.

3) Regarding chromosome individualization, the authors show that the percentage of "individualized chromosomes" is the same in mock and H1 depleted extracts (after dilution of the chromosome preparation before fixation). However, the dispersion of the chromosomes (processed for immunofluorescence without dilution) is clearly higher in the latter. What is driving the clustering? Interchromosomal entanglements? If so, shouldn´t the entanglements prevent chromosome individualization upon dilution?4) Related to point (3), the link between H1, condensin, Top2A and chromosome individualization is not clear. In Figure 6, chromosome individualization is measured by the separation of CENP-A foci, while in Figure 7, the authors examine surface area. These measurements may underlie quite distinct phenomena and so it is unclear what the different measurements in each of the conditions refer to. The authors need to clarify what these differences mean.

To better explain our observations and the apparent contradictions in the data, we added an additional figure discussing the two metrics for individualizations (Figure 5). We measured chromosome individualization in two contexts, with mechanical dispersal (Figure 6- Dilution assay) and in nocodazole treated extracts (Figure 7-Surface area).

Two lines of evidence demonstrate that significant interchromosomal links persist in mitosis and the TOP2A activity is needed to resolve these links. The first involves the clustering of chromosomes in the absence of a spindle (Figure 5B, 5C) which we quantify using the surface area of the chromosomal masses. The second is shown using the chromosome dilution assay (Figure 5A). In this assay, entropic forces, generated by dissolving the spindle matrix and reduced molecular crowding, disperse chromosomes and generate single chromosomes. The number of CENP-A foci per chromosome mass was a convenient way to quantitate chromosome clustering/individualization. Although the collapse of spindles due to the addition of nocodazole leads to chromosome clustering due to the accumulation of interchromosomal links (Figure 5A), the mechanical dispersion in the dilution assay can resolve the additional interchromosomal links (Figure 5D, 5E). We also then showed that the resolution of these interchromosomal links requires TOP2A activity during the dilution process (Figure 5D, 5E, figure supplement 1).

Due to the differences in the two assays, chromosome clustering observations by the dilution assay can differentiate between normal individualization in control (ΔIgG) and the hyperindividualization in ΔH1 extracts. However, the accumulation of interchromosomal links in ΔCAP-G extracts can be observed readily (Figure 6B, 6C). The surface area measurements assay measure individualization in nocodazole treated extracts (Figure 5A). Since no mechanical forces separate the chromosomes in this experiment, control extracts already appear to show maximal clustering (Figure 5B, 5C) and no differences between control and ΔCAP-G extracts seem to be observed. The hyper-individualization due to the loss of H1.8 mediated suppression of condensins and TOP2A is however observed as chromosome declustering.

5) On the same line, the authors write at the beginning of Discussion: "H1 (…) can also suppress chromosome dispersion that is driven by condensins and topo II that organize the long-range DNA folding". Two issues here:a) It is not clear which results in the paper actually connect long range DNA folding with the dispersion phenotype.b) The role of topoII in long range DNA folding is not tested in the paper.Maybe the contribution of topoII to chromosome length could be tested using the same scheme as in Figure 6G-H and the protocol for observing individualized chromosomes (dilution before fixation)

(a)We apologize for a grammatical mistake in this sentence, which should have read;

“Thus, the linker histone H1, which can promote local chromatin compaction through promoting nucleosome-nucleosome interaction, can also suppress chromosome dispersion that is driven by condensins and topo II, which organize the long-range DNA folding.”

Instead of “which”, we incorrectly connected the clauses with “that”. We meant to say that H1.8 plays a role in both long-distance genome folding and chromosome dispersion and this happens through its regulation of condensins and topo II. We do not wish to directly connect the long-range folding phenotype to the chromosome individualization phenotype. This is evident from our observations that shortened chromosomes with larger DNA loop layers in ΔH1ΔCAP-G extracts (Figure 3C, D, Figure 4D, E) are more dispersed than chromosomes in control extracts. Thus, although condensins and topo II may play a role in both processes, the processes appear to be regulated differently. To avoid the confusion, the rephrased sentence reads;

“Thus, the linker histone H1, which can promote local chromatin compaction through promoting nucleosome-nucleosome interaction, can also suppress chromosome dispersion and long-range DNA folding.”

(b) The data presented in Figure 3E, 3F show that chromosome length was determined solely by the condensin I levels on the chromatin, since condensin I co-depletion did notaffect the topo II overloading upon H1.8 depletion (Figure 7F). We also added Figure 3-figure supplement 2 where we show that partial TOP2A depletion leads to a small increase in chromosome length and that this does not rescue chromosome elongation in ΔH1 extracts. This is consistent with previous reports of the effect of TOP2A depletion or inhibition in vertebrate cells (Farr et al., 2014; Ladouceur et al., 2017; Nielsen et al., 2020; Samejima et al., 2012).

6) A previous study showed that depletion of condensin I results in complete loss of individualization of replicated chromosomes in *Xenopus* egg extracts (Shintomi and Hirano, Genes Dev 2011), whereas in the current study the authors see only a partial effect (Figure 5C). Could this mean that depletion is not complete (western bloting in Figure 5A suggests this might be the case)? Incomplete depletion would complicate the interpretation of the rescue experiment. Does the amount of protein in the extract and relate this to the extent of chromosomal condensation observed? The authors need to explain this discrepancy with previously published work.

As the reviewer notes, previous reports suggest that complete condensin I depletion fails to generate well-individualized chromosomes (Shintomi and Hirano 2011). We believe that the phenotype reported in Figure 6B, 6C is consistent with this above data. Since we mechanically separate chromosomes by a dilution step before fixing them, we believe that we can separate a small number of ‘fuzzy’ chromosomes from each nucleus in ΔCAP-G extracts. Since we observe only ~15-20% of the chromosomes are individualized even after mechanical dispersion, we believe that this represents a complete failure of chromosome individualization.

Please note that the interpretation of partial condensin depletion effect would have been complicated only if we had argued that apparent chromosome individualization in ΔH1ΔCAPG extracts is executed by a condensin-independent mechanism. Since additional condensin II co-depletion in triple depletion experiment (ΔH1ΔCAP-GΔCAP-D3) resulted in failed individualization, we instead argue that apparent chromosome individualization in ΔH1ΔCAPG extracts is due to increased chromatin-binding of condensin II (and likely due to residual condensin I). Although we may not eliminate the potential role of other proteins in promoting chromosome individualization in egg extract chromosomes, this possibility would not affect our conclusion that H1 suppresses chromosome individualization through limiting chromatinloading of condensin I and condensin II.

7) The authors propose that one of histone H1.8 role is to reduce chromosome length (a function achieved by antagonizing topo II/Condensin loading on chromosome). However, histone H1.8 is an embryonic linker histone and cells in early embryos are typically large and less dependent on chromosome size-reduction mechanisms. One would assume that chromosome size should be maximally reduced in small somatic cells and, consequently, shortening of chromosomes should be promoted by somatic linker histone H1 subtypes. In *Xenopus laevis*, histone H1.8 accumulates during oogenesis, persists until the mid-blastula transition, and is then replaced by somatic H1 subtypes. It seems like H1.8 disappears when its putative chromosome shortening function is most needed. The authors did not explain how their results are consistent with this biological context.

The reviewers are correct that one may assume that long chromosomes can be allowed in a large oocyte/egg. This is exactly why our discovery that the oocyte-specific H1.8 actively suppresses the actions of condensin is surprising and has important implications. Mitotic chromosome length is constrained not only by the cell size but also by the spindle size (Schubert and Oud 1997). Importantly, the spindle size in *Xenopus* embryos does not scale with the cell size during the early divisions (Wuhr et al., 2008). Functional significance of the upper limit of the spindle size remains speculative, but in the discussion, we suggested a potential role for limiting the spindle size to reduce the time of chromosome segregation/mitosis in rapid, synchronized cell divisions in frogs without cell cycle checkpoint controls. The longer the spindle is, the variations in duration for chromosome segregation may increase, making it difficult to maintain the cell division synchrony. Chromosomes prepared in H1.8 depleted extracts (ΔH1) are too long to fit into the mitotic spindles and this may lead to anaphase defects (Maresca, Freedman, and Heald 2005). H1.8 depletion also results in more fragile chromosomes, suggesting that condensin overloading due to loss of H1.8 may result in chromosome fragmentation due to spindle forces (Xiao et al., 2012).

Despite their smaller cell sizes, mitotic durations can be longer in somatic cells with checkpoint controls than in early embryonic cells with larger cell sizes. Indeed, it was suggested that the larger cell size in early embryos correlates with reduced mitotic checkpoint strength (Galli and Morgan 2016). Thus, it rather makes sense that having shorter mitotic chromosome length is more important in larger early embryonic cells that have reduced mitotic checkpoint strength. We have emphasized these points in our revised Discussion.

It is also possible that limiting the chromosome size may be a byproduct of preventing chromosome hyper-chromosome individualization by limiting the number of chromatin-bound condensins and TOP2A. As we pointed out in Discussion, clustering chromosomes during or prior to spindle formation by limiting chromosome individualization may be particularly important in a large oocyte to ensure that a bipolar spindle assembles on the whole set of chromosomes. Pronuclear clustering in human and bovine oocytes has also recently been shown to be important to ensure efficient chromosome segregation (Cavazza et al., 2021). Previous studies have also shown that, when bound to chromatin, somatic linker histones can rescue the chromosome elongation observed in H1.8 depleted extracts (Freedman and Heald 2010). However, exogenously provided somatic linker histones are sequestered by importin β during mitosis and are thus excluded largely from mitotic chromatin and somatic linker histones can only rescue loss of H1.8 upon release from this sequestration (Freedman and Heald 2010). This suggests that H1.8 plays a specialized role in oocyte and early embryo mitoses to restrict the chromosome length and individualization in these cells. Somatic linker histones may also play a similar role in the chromosome shortening observed in post MBT embryos as importin sequestration is weaker in smaller cells (Freedman and Heald 2010).

Reviewer #1 (Recommendations for the authors):1. Is there a threshold amount of condensin I that needs to be associated with chromosomes for "normal" length. Can the authors partially deplete histone H1 and see a correlation between the amount of condensin and chromosome length?

In Figure 3E, F, by co-depletion of H1.8 and condensin I, we were able to generate a broad range of chromosome-bound condensin I levels. The chromosome lengths are highly correlated with the amount of chromosome-bound condensin I in this range. Since condensin I is essential for chromosome individualization, a process that is needed for chromosome length measurement, it might not be simple to address the question of whether a threshold level of condensin is required for “normal” chromosome length. Based on Figure 3E, F and Hi-C data, we would assume that chromosome length is linearly correlated with the chromosome-bound levels of condensin I when there is enough condensin I to make chromosome individualized.

2. Related to 2. If chromosome individualization is affected in H1 depletion, the authors should be able to observe inter-chromosomal interactions in their Hi-C dataset. Can this be quantified for the different conditions?

This is an interesting question, and we tried to monitor inter-chromosomal interactions by Hi-C. Unfortunately, the data were difficult to interpret. As the ligation step was performed in situ as described previously (Belaghzal, Dekker, and Gibcus 2017), the lack of a cellular membrane in pelleted mitotic nuclei from *Xenopus* egg extracts likely resulted in increased inter-nuclear trans contacts. Although this data still captures the mitotic chromosome structure through cis contacts, we observe a high incidence of trans contacts that may be artifactual. While it may be possible to reduce internuclear trans contacts by modifying the dilution procedure, this requires substantial efforts and is thus beyond the scope of the current manuscript.

Reviewer #2 (Recommendations for the authors):This is a very nice study from a laboratory that has contributed several important papers to the field of mitosis over the years. Experiments were well designed and carefully performed for the most part (with the exception of DNA/nucleosome binding experiments with purified Condensins). The flow of experiments in this manuscript is logical. Overall, I would say this manuscript addresses an important biological question, and this topic has obvious implications for development and genome stability. The manuscript would however benefit from toning down several overinterpretations and would also be improved by providing a more compelling rationale to justify a role for an early embryonic histone in the reduction of chromosome length.

Regarding the functional implications of our findings in oocytes and early embrios, please note our response to Essential Point #7, where we explain why it rather makes sense to keep chromosomes short and suppress hyper chromosome individualization prior to anaphase in a large egg/oocyte.

1 – All experiments presented in this study should be performed a minimum of 3 times as this represents a basic requirement for reproducibility in biomedical research. This includes all Hi-C experiments (two replicates are reported) and some nucleosome-binding experiments (Figure 2—figure supplement 3).

We disagree with an idea that all experiments must be repeated a minimum of 3 times in biomedical research before sharing the results with public; it is not an established rule in the cell biology field (or eLife publication policy), as far as we are concerned. We employed a variety of complementary methods (quantitative microscopy, biochemistry and Hi-C) to make conclusive remarks. We do not see that the level of confidence significantly increases by enforcing every experiment to be done 3 times, statistically speaking. Ultimately, reproducibility must be independently tested by other researchers. However, we have repeated the mono-nucleosome-binding experiment (Figure 2—figure supplement 4), and further strengthened our original conclusion.

2 – There is a missing reference in the following sentence on page 4: Since Condensins prefer to bind nucleosome-free DNA (Kong et al. 2020; Zierhut et al. 2014; Shintomi et al. 2017; Toselli‐Mollereau et al. 2016). This observation was originally made by Piazza et al. 2014, Nat Struct Mol Biol.

We thank the reviewer for pointing out this. We will include this reference.

Reviewer #3 (Recommendations for the authors):It could be very useful to present a table in the Discussion with a summary of:1 – the relative amounts of condensins I and II and topo II on DNA , taken from figures 5D, 5E, and Figure 6_supplement1 C2 –the phenotype observed in terms of chromosome length, chromosome individualization, and clustering, under the different conditions tested (control, δ H1, deltaH1deltaG, deltaH1deltaD3 and deltaH1deltaGdeltaD3).This would help understand the co-dependencies in recruitment of the factors under study and the correlation between the different phenotypes.

Thank you for the great idea. We added this table as a Table 1.

References

Belaghzal, Houda, Job Dekker, and Johan H. Gibcus. 2017. “Hi-C 2. 0 : An Optimized Hi-C Procedure for High-Resolution Genome-Wide Mapping of Chromosome Conformation.” Methods 123: 56–65. https://doi.org/10.1016/j.ymeth.2017.04.004.

Cavazza, Tommaso, Yuko Takeda, Antonio Z Politi, Magomet Aushev, Patrick Aldag, Clara Baker, Meenakshi Choudhary, et al. 2021. “Parental Genome Unification Is Highly Error-Prone in Mammalian Embryos.” Cell, May, 1–18. https://doi.org/10.1016/j.cell.2021.04.013.

Farr, Christine J., Melissa Antoniou-Kourounioti, Michael L. Mimmack, Arsen Volkov, and Andrew C.G. Porter. 2014. “The α Isoform of Topoisomerase II Is Required for Hypercompaction of Mitotic Chromosomes in Human Cells.” Nucleic Acids Research 42 (7): 4414–26. https://doi.org/10.1093/nar/gku076.

Freedman, Benjamin S., and Rebecca Heald. 2010. “Functional Comparison of H1 Histones in *Xenopus* Reveals Isoform-Specific Regulation by Cdk1 and RanGTP.” Current Biology 20 (11): 1048–52. https://doi.org/10.1016/j.cub.2010.04.025.

Freedman, Benjamin S., Kelly E. Miller, and Rebecca Heald. 2010. “*Xenopus* Egg Extracts Increase Dynamics of Histone H1 on Sperm Chromatin.” PLoS ONE 5 (9): 1–10. https://doi.org/10.1371/journal.pone.0013111.

Galli, Matilde, and David O. Morgan. 2016. “Cell Size Determines the Strength of the Spindle Assembly Checkpoint during Embryonic Development.” Developmental Cell 36 (3): 344–52. https://doi.org/10.1016/j.devcel.2016.01.003.

Ganji, Mahipal, Indra A. Shaltiel, Shveta Bisht, Eugene Kim, Ana Kalichava, Christian H. Haering, and Cees Dekker. 2018. “Real-Time Imaging of DNA Loop Extrusion by Condensin.” Science 360 (6384): 102–5. https://doi.org/10.1126/science.aar7831.

Kimura, Keiji, Olivier Cuvier, and Tatsuya Hirano. 2001. “Chromosome Condensation by a Human Condensin Complex in *Xenopus* Egg Extracts.” The Journal of Biological Chemistry 276 (8): 5417–20. https://doi.org/10.1074/jbc.C000873200.

Kimura, Keiji, and Tatsuya Hirano. 1997. “ATP-Dependent Positive Supercoiling of DNA by 13S Condensin: A Biochemical Implication for Chromosome Condensation.” Cell 90 (4): 625–34. https://doi.org/10.1016/S0092-8674(00)80524-3.

Kong, Muwen, Erin E. Cutts, Dongqing Pan, Fabienne Beuron, Thangavelu Kaliyappan, Chaoyou Xue, Edward P. Morris, Andrea Musacchio, Alessandro Vannini, and Eric C. Greene. 2020. “Human Condensin I and II Drive Extensive ATP-Dependent Compaction of Nucleosome-Bound DNA.” Molecular Cell, May, 683540. https://doi.org/10.1016/j.molcel.2020.04.026.

Ladouceur, Anne-Marie, Rajesh Ranjan, Lydia Smith, Tanner Fadero, Jennifer Heppert, Bob Goldstein, Amy Shaub Maddox, and Paul S. Maddox. 2017. “CENP-A and Topoisomerase-II Antagonistically Affect Chromosome Length.” The Journal of Cell Biology 216 (9): jcb.201608084. https://doi.org/10.1083/jcb.201608084.

Maresca, Thomas J., Benjamin S. Freedman, and Rebecca Heald. 2005. “Histone H1 Is Essential for Mitotic Chromosome Architecture and Segregation in *Xenopus laevis* Egg Extracts.” Journal of Cell Biology 169 (6): 859–69. https://doi.org/10.1083/jcb.200503031.

Nielsen, Christian F., Tao Zhang, Marin Barisic, Paul Kalitsis, and Damien F. Hudson. 2020. “Topoisomerase IIa Is Essential for Maintenance of Mitotic Chromosome Structure.” Proceedings of the National Academy of Sciences of the United States of America 117 (22). https://doi.org/10.1073/pnas.2001760117.

Samejima, Kumiko, Itaru Samejima, Paola Vagnarelli, Hiromi Ogawa, Giulia Vargiu, David A. Kelly, Flavia de Lima Alves, et al. 2012. “Mitotic Chromosomes Are Compacted Laterally by KIF4 and Condensin and Axially by Topoisomerase IIα.” Journal of Cell Biology 199 (5): 755–70. https://doi.org/10.1083/jcb.201202155.

Schubert, I., and J. L. Oud. 1997. “There Is an Upper Limit of Chromosome Size for Normal Development of an Organism.” Cell 88 (4): 515–20. https://doi.org/10.1016/S0092-8674(00)81891-7.

Shintomi, Keishi, and Tatsuya Hirano. 2011. “The Relative Ratio of Condensin I to II Determines Chromosome Shapes.” Genes and Development 25 (14): 1464–69. https://doi.org/10.1101/gad.2060311.

Woodcock, Christopher L., Arthur I. Skoultchi, and Yuhong Fan. 2006. “Role of Linker Histone in Chromatin Structure and Function: H1 Stoichiometry and Nucleosome Repeat Length.” Chromosome Research 14 (1): 17–25. https://doi.org/10.1007/s10577-005-1024-3.

Wuhr, Martin, Yao Chen, Sophie Dumont, Aaron C. Groen, Daniel J. Needleman, Adrian Salic, and Timothy J. Mitchison. 2008. “Evidence for an Upper Limit to Mitotic Spindle Length.” Current Biology 18 (16): 1256–61. https://doi.org/10.1016/j.cub.2008.07.092.

Xiao, B., Benjamin S. Freedman, K. E. Miller, Rebecca Heald, and John F. Marko. 2012. “Histone H1 Compacts DNA under Force and during Chromatin Assembly.” Molecular Biology of the Cell 23 (24): 4864–71. https://doi.org/10.1091/mbc.E12-07-0518.

[Editors' note: further revisions were suggested prior to acceptance, as described below.]

You will see below that Reviewer #2 has two outstanding issues. Please address these as follows:1. Please add a sentence to the Results section for the condensin binding assays to qualify that SMC subunits may be retained on the chromatin beads more efficiently than the other condensin subunits. You may also want to speculate why this is the case, as in your rebuttal letter.

We revised the corresponding sentences to read:

“We noticed that the subunit stoichiometry of condensin I and condensin II was altered in the bead fraction from that in the input. Since the condensin complexes were intact at the assay conditions (Figure 2—figure supplement 1), we suspect that non-SMC subunits of condensins were less stable on the nucleosome array than SMC subunits, while H1.8 reduced binding of all condensin subunits to the nucleosome array.”

2. While three repetitions of every experiment are not insisted on, it is important that the statistical analysis used is appropriate and clearly reported throughout the manuscript. In a number of places in the manuscript, t tests appear to have been carried out where n=2. If/where this is the case, please reanalyse the data using the appropriate statistical test and state this clearly in the figure legend. Where key experiments fail to show significance using these criteria, an additional repeat may be appropriate. For example, the legend of Figure 2 states "The statistical significance of the changes in each immunofluorescence experiment was assessed by a Mann Whitney U-test and the p-values shown in D, E and G are calculated by an unpaired Student's t-test of the aggregate medians (immunofluorescence) or blot intensities of each single experiment." A t-test would not be appropriate for Figure 2D where it appears that two immunofluorescence experiments are presented, if this is indeed what is shown. Please also refer to the comment by reviewer #2.

We assume that the editor meant Figure 1, not Figure 2.

In general (Figures 1D, 3D, 6D, 7E, 7F), the statistical analysis of superplots with data from multiple experiments was performed as described (Lord et al., 2020). The t-test comparing just the medians of the samples is the most stringent significance test that ignores the confidence generated by the technical replicates in each sample. We also note that the t-test is still valid for n=2, although it reduces statistical power (Naegle, Gough, and Yaffe 2015). This may result in false negatives, where smaller differences between samples are likely to be deemed non-significant. However, since the changes we observed were substantial, we observed significantly low p values with the reduced power of n=2 experiments. However, as the reviewer notes, the t-test with the reduced n value may lead to erroneous conclusions. As suggested, we changed the analysis in those experiments where the experiment was only repeated twice to a non-parametric test (Figure 1G, 6E). We also added an additional experiment for Figure 1D, where we now applied Student’s t-test. For Figure 6B, 6C, although similar experiments were repeated multiple times, the CENP-A metric was only performed twice. To supplement this, we now include additional data performed independently thrice, where chromosome individualization was quantified based on manual chromosome morphological classification (Figure 6—figure supplement 1C).

We also wish to stress that none of the conclusions of the paper rely on the statistical significance of any single measurement. The functional consequences of condensin and TOP2A suppression by H1.8 were shown by several different orthogonal experiments.

In addition, please note suggestions for textual changes from Reviewer #1.Reviewer #1 (Recommendations for the authors):The authors have addressed all of the essential revisions. In addition they have provided convincing responses to all reviewers comments. There are a few places where the text could be clarified:

We appreciate these specific suggestions. We have clarified several of the statements and added more context where it was necessary.

Line 185. Please explain the Q loop mutant.

The revised text reads, “the ATP-binding deficient Q-loop mutant of condensin I”. Please note that this was also introduced at line 164.

Line 195 "Although SMC2 and SMC4 appeared to be better retained on nucleosome beads than the non-SMC subunits"

This section now reads, “We noticed that the subunit stoichiometry of condensin I and condensin II was altered in the bead fraction from that in the input. Since the condensin complexes are intact at the assay conditions (Figure 2—figure supplement 1), we suspect that non-SMC subunits of condensins were less stable on the nucleosome array than SMC subunits, while H1.8 reduced binding of all condensin subunits to the nucleosome array.”

Line 318 "This is consistent with the shorter chromosomes in DH1DCAP-G1 extracts as compared to control…"

The revised sentence reads:

“Further supporting the idea that the layer size anticorrelates with the chromosome length, chromosomes in ΔH1ΔCAP-G extracts were indeed shorter than those in control ΔIgG extracts (Figure 3D, 3F).”

Line 362: Please explain ICRF-193.

The revised sentence reads, “Indeed, when ICRF-193, the drug that inhibits topo II-dependent catenation/decatenation without leaving double-strand DNA breaks (Ikegami, Ishida, and Andoh 1991), was added to egg extracts at the mitotic entry, chromosome individualization was completely blocked, forming large chromosome clusters after extract dilution (Figure 5—figure supplement 1B-D, ICRF-50 min).”

Line 368: Suggest: To quantify the clustering, we stained the coverslips for CENP-A and used the CENP-A foci (CENP-A doublet is counted as one focus) to measure the number of chromosomes in each chromosome mass. The fraction of chromosomes was measured in small clusters (<4 CENP-A foci, since a few chromosomes can colocalizae stochastically).

As we realized that we never introduced CENP-A, we revised this section to better explain this method as below:

“To quantify the clustering of unresolved chromosomes, we stained the coverslips for CENP-A, the centromere-specific histone H3 variant, which marks a centromere locus per chromosome (Edwards and Murray 2005). While each fully individualized chromosome showed one CENP-A focus (CENP-A doublet, representing centromeres of a paired sister chromatids, is counted as one focus), multiple CENP-A foci were observed in a cluster of chromosomes. Since even fully individualized chromosomes stochastically interacted during chromosome dispersion, we redefined the chromosome cluster here for the chromosome mass containing four or more CENP-A foci (Figure 5D, 5E, Figure 5—figure supplement 1E). Although chromosomes from each nucleus tightly compact into a ball-like mass in nocodazole treated metaphase extracts (Figure 5B), they individualized normally by dispersion upon extract dilution (Figure 5D, 5E, figure supplement 1E).”

Line 383 "better" can be deleted here since the assay has already been introduced.

We changed the sentence as advised.

Line 421 "showed reduced clustering compared to.."

We changed the sentence as advised.

Line 427 Suggest replacing "background" with "extracts"

We followed the advice.

Line 428 "TOP2A depletion in H1.8 depleted extracts also did not reduce the chromosome spreading even though TOP2A levels on these chromosomes became comparable to that of the control".

We incorporated this suggestion.

Reviewer #2 (Recommendations for the authors):The revised manuscript has been improved by the authors but two issues raised in previous reviews have not been addressed in the latest version of the manuscript.

We respectfully disagree with the assessment of the reviewer.

1 – Condensin stoichiometry: Experiments shown in Figure 2 and associated supplemental figures use Condensin complexes with non-stoichiometric subunit composition. The authors recognize this fact in their rebuttal letter, stating that the "stoichiometry of condensin subunits seems to be changed in DNA-bound fraction in DNA-beads pull-down experiments…" The issue is not convincingly resolved by the mass photometry experiment shown in Figure 2—figure supplement 1 because this experiment does not fully recapitulate conditions of the DNA binding reactions.The authors suggest problems with stoichiometry are not relevant because Condensin complexes used in their experiments were "purified as reported previously (Kong et al., 2020)" and "are similar to those seen in previous reports in both recombinant yeast condensin and immunoprecipitated *Xenopus* and human condensins (Kong et al., 2020; Ganji et al., 2018; Kimura, Cuvier, and Hirano 2001; Kimura and Hirano 1997)." Examination of Condensin purification gels from Kong et al. shows that the complexes purified by these authors are actually stoichiometric (see Figure 1D in PMID 32445620). The same conclusion can be reached for the yeast complexes purified by the Häring group (see Figure 1C in PMID 28882993). The other two studies cited by the authors (Kimura, Cuvier, and Hirano 2001; Kimura and Hirano 1997) use silver staining to label Condensin subunits, a staining method that most biochemists recognize as non-linear and inappropriate to compare the abundance of different proteins. The contention that subunit imbalances seen in the Condensin complexes used by Choppakatla and colleagues are normal and/or reflect imbalances seen in other purified Condensin complexes cannot be verified in the published literature.The DNA/nucleosome binding behavior attributed to Condensin is this study is likely reflective of the SMC subcomplex (CAP-E/C without CAP-G/H/D2), not the genuine Condensin 1 enzyme. As a consequence, most or all the DNA/nucleosome array binding experiments need to be repeated with stoichiometric Condensin complexes to provide valid conclusions on the DNA binding behavior of the entire/native enzyme.

The mass photometry experiments shown in Figure 2—figure supplement 1 were performed in the assay buffers and at assay conditions (Room temperature incubation). It is not immediately obvious to us why this does not completely recapitulate the experimental conditions in the binding reactions in Figure 2. Since mass photometry is a more direct measurement of the complex stoichiometry than the Coomassie staining on the gels, we are confident that the input condensin complexes are stoichiometric in nature. Please note that the mass photometry analysis was done on the aliquots from the same lot of purified condensin complexes used in all experiments shown in this manuscript. Although the stoichiometry in the condensin complexes bound to the beads appears to be different, we feel that this may be due to a differential elution from the chromatin beads during the washing step. Supporting this possibility, the electrophoretic mobility shift assays (EMSA) performed in solution also recapitulate the H1.8 mediated inhibition of condensin binding (Figure 2—figure supplement 4). Importantly, we verified that the purified condensin I complex can support condensin I-depletion phenotypes in *Xenopus* egg extracts (Figure 2—figure supplement 2A). This is an ultimate verification that our purified condensin I is fully functional.

2 – Several experiments presented in the revised manuscript have been performed as duplicates, and on one occasion (Figure 3E-F) the results presented come from a single experiment. This reviewer argued previously that all experiments in this study should be performed a minimum of 3 independent times. The authors stated in their rebuttal letter that they "do not see that the level of confidence significantly increases by enforcing every experiment to be done 3 times, statistically speaking." I disagree with this statement. There is ample evidence to show that statistical analyses are less reliable with sample sizes of 2. Many textbooks /reference papers in statistics support this view and, as an example, I would refer the authors to "A biologist's guide to statistical thinking and analysis." The authors are correct, though, in their statement that eLife has no explicit publication policy on the minimal number of independent replicates required for publication.The insistence of the authors to use two independent datasets/experiments for most figures has statistical implications. For example, on some occasions, the authors use a t-test to compare two groups, which is a parametric test. On other occasions, the authors use a Mann Whitney U test, which is a non-parametric test. Parametric testing assumes that the data are normally distributed. With n=2, it is expected that the data would not pass a normality test, hence a Mann-Whitney U test is likely the only appropriate option for most of the figures presented in the manuscript. Separate from these points, when comparing multiple groups on a graph, a One-Way ANOVA is more appropriate than performing multiple independent t-tests. Overall, the authors need to justify their rationale for the choice of statistical analyses performed throughout the manuscript, especially when using only n=2. In many cases, performing an extra experiment to bring datasets to n=3 would likely alleviate the concerns expressed above.

In general (Figures 1D, 3D, 6D, 7E, 7F), the statistical analysis of superplots with data from multiple experiments was performed as described here (Lord et al., 2020). The t-test comparing just the medians of the samples is the most stringent significance test that ignores the confidence generated by the technical replicates in each sample. We also note that the t-test is still valid for n=2, although it loses statistical power (Naegle, Gough, and Yaffe 2015). This may result in false negatives, where smaller differences between samples are likely to be deemed non-significant. However, since the changes we observed were substantial, we observed significantly low p values with the reduced power of n=2 experiments. However, as the reviewer notes, the t-test with the reduced n value may lead to erroneous conclusions. As suggested, we changed the analysis in those experiments where the experiment was only repeated twice to a non-parametric test (Figure 1G, 6E). We also added an additional experiment for Figure 1D, where we now applied Student’s t-test. For Figure 6B, 6C, although similar experiments were repeated multiple times, the CENP-A metric was only performed twice. To supplement this, we now include additional data performed independently thrice, where chromosome individualization was quantified based on chromosome morphological classification (Figure 6- figure supplement 1C).

Regarding the experiment shown in Figure 3E-F, the result was reproduced in a biological replicate (Figure 3- figure supplement 1C,D), as we had reported in the previous revision.

We also wish to stress that none of the conclusions of the paper rely on the statistical significance of any single measurement. The functional consequences of condensin and TOP2A suppression by H1.8 were shown by several different orthogonal experiments. Collectively, we believe that it is highly unlikely that addition or even subtraction of an experiment would debunk our major conclusions described in this paper. Thus, we stand by the scientific rigor of the manuscript.

References

Edwards, Nathaniel S, and Andrew W Murray. 2005. “Identification of *Xenopus* CENP-A and an Associated Centromeric DNA Repeat.” Molecular Biology of the Cell 16 (4): 1800–1810. https://doi.org/10.1091/mbc.e04-09-0788.

Ikegami, Yoji, Ryoji Ishida, and Toshiwo Andoh. 1991. “Inhibition of Topoisomerase II by Antitumor Agents Bis(2,6-Dioxopiperazine) Derivatives.” Cancer Research 51 (18): 4903–8.

Lord, Samuel J, Katrina B Velle, R Dyche Mullins, and Lillian K. Fritz-Laylin. 2020. “SuperPlots: Communicating Reproducibility and Variability in Cell Biology.” Journal of Cell Biology 219 (6). https://doi.org/10.1083/jcb.202001064.

Naegle, Kristen, Nancy R Gough, and Michael B Yaffe. 2015. “Criteria for Biological Reproducibility: What Does ‘n ‘ Mean?” Science Signaling 8 (371): fs7–fs7. https://doi.org/10.1126/scisignal.aab1125.